# Plasmepsin II–III copy number accounts for bimodal piperaquine resistance among Cambodian Plasmodium falciparum

Selina Bopp [1], Pamela Magistrado[1], Wesley Wong[1], Stephen F. Schaffner [1,2], Angana Mukherjee[1], Pharath Lim [3], Mehul Dhorda[4,5,6], Chanaki Amaratunga[3], Charles J. Woodrow[5], Elizabeth A. Ashley[6,7], Nicholas J. White[5,7], Arjen M. Dondorp[5,7], Rick M. Fairhurst[3], Frederic Ariey[8], Didier Menard[9,10,11], Dyann F. Wirth[1,2] & Sarah K. Volkman[1,2,12]

Multidrug resistant *Plasmodium falciparum* in Southeast Asia endangers regional malaria elimination and threatens to spread to other malaria endemic areas. Understanding mechanisms of piperaquine (PPQ) resistance is crucial for tracking its emergence and spread, and to develop effective strategies for overcoming it. Here we analyze a mechanism of PPQ resistance in Cambodian parasites. Isolates exhibit a bimodal dose–response curve when exposed to PPQ, with the area under the curve quantifying their survival in vitro. Increased copy number for *plasmepsin II* and *plasmepsin III* appears to explain enhanced survival when exposed to PPQ in most, but not all cases. A panel of isogenic subclones reinforces the importance of *plasmepsin II–III* copy number to enhanced PPQ survival. We conjecture that factors producing increased parasite survival under PPQ exposure in vitro may drive clinical PPQ failures in the field.

---

[1] Harvard T.H. Chan School of Public Health, Boston, MA 02115, USA. [2] The Broad Institute of MIT and Harvard, Cambridge, MA 02142, USA. [3] National Institutes of Health, Rockville, MD 20892, USA. [4] Worldwide Antimalarial Resistance Network, Bangkok 10400, Thailand. [5] Mahidol-Oxford Tropical Medicine Research Unit, Bangkok 10400, Thailand. [6] Myanmar-Oxford Clinical Research Unit, Yangon, Myanmar. [7] Centre for Tropical Medicine and Global Health, University of Oxford, Oxford OX3 7FZ, UK. [8] Institut Cochin, INSERM U:1016, Parasitology-Mycology Unit, Cochin Hospital Paris Descartes University, Paris 75014, France. [9] Biology of Host-Parasite Interactions Unit, Institut Pasteur, Paris 75015, France. [10] CNRS, ERL 9195, Paris 75794, France. [11] INSERM, Unit U1201, Paris 75015, France. [12] Simmons College, Boston, MA 02115, USA. Correspondence and requests for materials should be addressed to S.K.V. (email: svolkman@hsph.harvard.edu)

D rug resistance is a major threat to global efforts to control, eliminate and eradicate malaria. Artemisinin (ART) and related compounds are currently the main class of antimalarial drugs; they are used in combination with partner drugs to forestall the emergence and spread of resistance. Accordingly, ART combination therapies (ACTs) provide first-line drug treatment for uncomplicated *Plasmodium falciparum* infection[1]. Recent emergence and spread of increasingly ART-resistant parasites is evident in Southeast Asia (SEA), characterized by increased parasite clearance half-life in patients[2,3] and increased survival in vitro when assessed by the ring-stage survival assay[4]. Mutations in the *Pfkelch13* locus are associated with[5] and confer ART resistance[6,7]. Partner drug resistance is also evident; including emergence of piperaquine (PPQ) resistance in Cambodia and Viet Nam[8–10], and mefloquine (MEF) resistance on the Thailand–Myanmar border[11,12]. Recently, a dominant parasite lineage from Cambodia that harbors both ART and PPQ resistance was shown to be widespread[13], heightening concern about highly resistant parasites that threaten dihydroartemisinin (DHA)–PPQ-based interventions such as mass drug administration.

Despite clinical evidence that DHA–PPQ treatment failures have increased in SEA, detection of PPQ resistance has been challenging and time consuming, with conventional in vitro dose–response assays frequently yielding non-interpretable data[14,15]. A PPQ survival assay (PSA) was developed to identify PPQ resistance and treatment failures more easily and reliably[16], but remains time intensive owing to manual slide counting requirements. Thus, a better phenotypic test would facilitate tracking of PPQ resistance. Key genetic markers associated with PPQ resistance include copy number variations (CNV) in both the *P. falciparum* multidrug resistance gene 1 (*pfmdr1*, resistance associated with decreased copy number)[17,18] and *plasmepsin II* and *plasmepsin III* (*plasmepsin II–III*, resistance associated with increased copy number)[18–20]. Other loci that potentially have a role in PPQ resistance have been identified[17,18,21–23], but the biological mechanisms underlying PPQ resistance are not well understood.

To better understand the biology of PPQ resistance, we analyzed a set of culture-adapted Cambodian parasites from the Tracking Resistance to Artemisinin Collaboration (TRAC), collected in Pursat and Pailin in 2011[24] when the first cases of

recrudescence were reported[19]. We tested 37 culture-adapted parasites from this collection of 157 parasites[25], and found differential response to increasing concentrations of PPQ, with several isolates exhibiting a bimodal dose–response under high levels of drug. We interrogated the correlation between this bimodal response and genetic loci including copy numbers for several genes including *plasmepsin II–III* and *pfmdr1*. To disentangle the contributions of these two loci to PPQ resistance in this population, we studied a panel of isogenic parasite isolates with a single *pfmdr1* copy but variable *plasmepsin II–III* CNVs, and found that *plasmepsin II–III* is the major driver of PPQ resistance. However, evidence of discordance between PPQ resistance and *plasmepsin II–III* CNV suggests that additional genetic variants or expression profiles are involved in PPQ resistance among these Cambodian parasites. Characterizing the survival of cultured parasites exposed to PPQ reveals changes that inform possible mechanisms of clinical PPQ resistance evident in Cambodia. We thus describe the biological response of these parasites to PPQ, and investigate potential involvement of genetic loci to advance our knowledge about emerging PPQ resistance in Cambodia.

## Results

**Area under the curve (AUC) as a measure of PPQ resistance.** One challenge to investigating PPQ resistance is defining the in vitro drug resistance phenotype for PPQ. Conventional drug susceptibility testing over 72 h to measure the half-maximal effective concentration ($EC_{50}$) resulted in incomplete parasite killing, with several parasite isolates surviving the highest drug concentration used. Using the PSA was not optimal since it was very labor-intensive and prone to microscopist bias. To gain better understanding of the biological response of parasites to PPQ and define PPQ resistance within this Cambodian parasite population, culture-adapted clinical isolates were subjected to a modified dose–response approach that resulted in complete parasite killing. To achieve complete parasite killing for all isolates, the starting PPQ concentration was increased 100-fold (from 0.5 to 50 μM) and the dilution series extended from 12 to 24 points. Highly synchronized ring-stage parasites (0–6 h) were exposed to these conditions for 72 h. Under these conditions, complete killing of all parasites at the highest drug concentration was achieved (Fig. 1a, Supplementary Fig. 1).

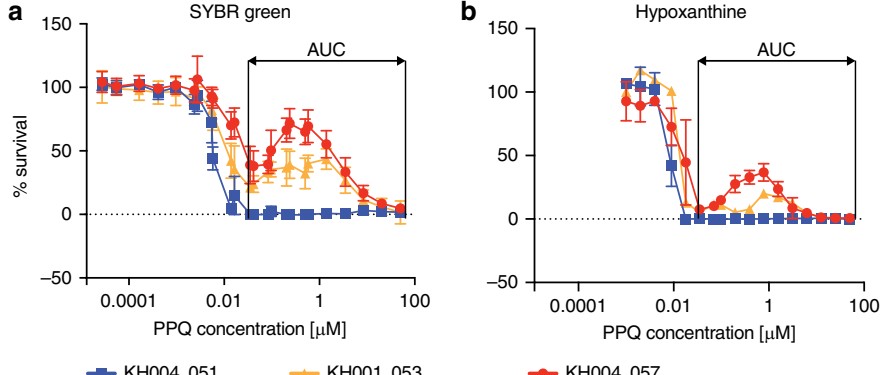

**Fig. 1** Bimodal distribution of parasite response to PPQ exposure. The Area Under the Curve (AUC) value was determined using the local minima as boundaries to describe the parasite response to PPQ. Increasing the starting concentration and number of data points for the conventional SYBR Green dose–response curve **a** provided a better curve representation and brought percent survival to zero at higher concentrations. Bimodal growth was observed with increasing PPQ concentrations for a subset of parasites. Representative PPQ-resistant parasites KH004_057 (red) and KH001_053 (orange) and a PPQ-sensitive parasite KH004_051 (blue) are shown. Similar data for all 37 isolates tested can be found in Supplementary Fig. 1. Hypoxanthine assays **b** confirmed that these dose–response results were derived from viable parasites. All drug assays were done in triplicate and repeated two times, with the mean and s.d. shown for each concentration

Investigation of responses to PPQ exposure revealed that several isolates showed a bimodal response to PPQ rather than a classical sigmoidal dose–response curve typical for antimalarial drugs (Fig. 1a, Supplementary Fig. 1). Although parasites were killed at the same rate between 1 and 10 nM PPQ concentrations, there was a difference in how well parasites could survive under higher drug concentrations (0.1 μM–10μM), resulting in a second peak of survival for a subset of the parasites (Fig. 1a, Supplementary Fig. 1). The first half of the second dose–response peak overlaps with PPQ plasma concentrations between 30 and 300 ng/ml (or 56–560 nM) found in patients after 3 days of PPQ treatment[26]. We confirmed that parasites exhibiting this high secondary peak were also viable at high PPQ concentrations assessed by [3H]-hypoxanthine incorporation assays (Fig. 1b), which requires the parasite to actively synthesize DNA. As it was impossible to determine a conventional $EC_{50}$ value for PPQ response for the bimodal curve, we calculated the AUC to quantify the parasite survival response to PPQ. AUC was calculated within the region of curve bounded by drug concentrations of 0.1 μM and 30 μM, defined by the average location of local minima flanking the second dose–response peak in samples displaying a bimodal dose–response. Thus, the AUC quantified the PPQ resistance phenotype for each of these Cambodian parasites.

We observed a range of AUC values (Supplementary Fig. 1) with a distribution that suggested three discrete groups (Supplementary Fig. 1b), with the first breakpoint at ~35 and the second at ~100. We compared AUC values for nine isolates with results obtained using the published PSA[16] to establish a resistance cutoff (Supplementary Fig. 2). In contrast to the dose–response assay, the PSA exposes 0–3 h ring-stage parasites to 200 nM PPQ for 48 h before determining parasitemia by microscopy. The relative growth of the drug-treated parasite compared with non-drug-treated control parasite culture after 72 h was used to measure PPQ response with isolates demonstrating relative growth of >10% considered PPQ-resistant[16]. Among

these nine representative isolates, we found that PSA survival rates between 0 and 30% correlated well with AUC values (Spearman $r = 0.85$, $p = 0.0061$; Supplementary Fig. 2). Isolates with AUC > 100 were well above the >10% resistance cutoff defined in the PSA and we consider them PPQ resistant (PPQR). Isolates with AUC < 35 were all below the 10% cutoff and considered sensitive (PPQS).

**PPQ resistance is present on different genetic backgrounds.** The *P. falciparum* population in SEA is highly structured[27,28], with only a few distinct subpopulations or clades. Our parasite isolates from Pursat and Pailin fall within two clades (KH1 and KH4), both of which harbor PPQ resistance[27,28]. To test whether PPQ resistance occurred on a similar genetic background, we leveraged sequencing data for 157 parasites[29] and carried out an identity by descent (IBD) analysis for relatedness[30,31]. Multiple highly related clusters (i.e., those that share >90% of their genome) were evident, with the largest cluster containing parasites exhibiting both intermediate (AUC = 35–100) and high levels (AUC > 100) of PPQ responses. However, multiple PPQR parasites were genetically unrelated (Fig. 2). As ART resistance has contributed to this population substructure, we investigated the distribution of PPQ response among parasites either harboring or lacking mutations in *Pfkelch13* that confer ART resistance (Supplementary Fig. 3). PPQR isolates contained either a C580Y or Y493H mutation, whereas PPQS isolates were either wild-type or had one of several different *PfKelch13* mutations (C580Y, E270K, R539T, I543T, D584V, or H719N). PPQR isolates with intermediate phenotypes (AUC = 35–100) were either wild-type or had a C580Y mutation. The only *PfKelch13* mutation exclusively found in PPQR parasites was Y493H; and, parasites with this mutation were distributed both among highly related and unrelated subpopulations. This Y493H allele was represented in 6 of the 157 isolates overall, two of which were phenotyped for PPQ response[25].

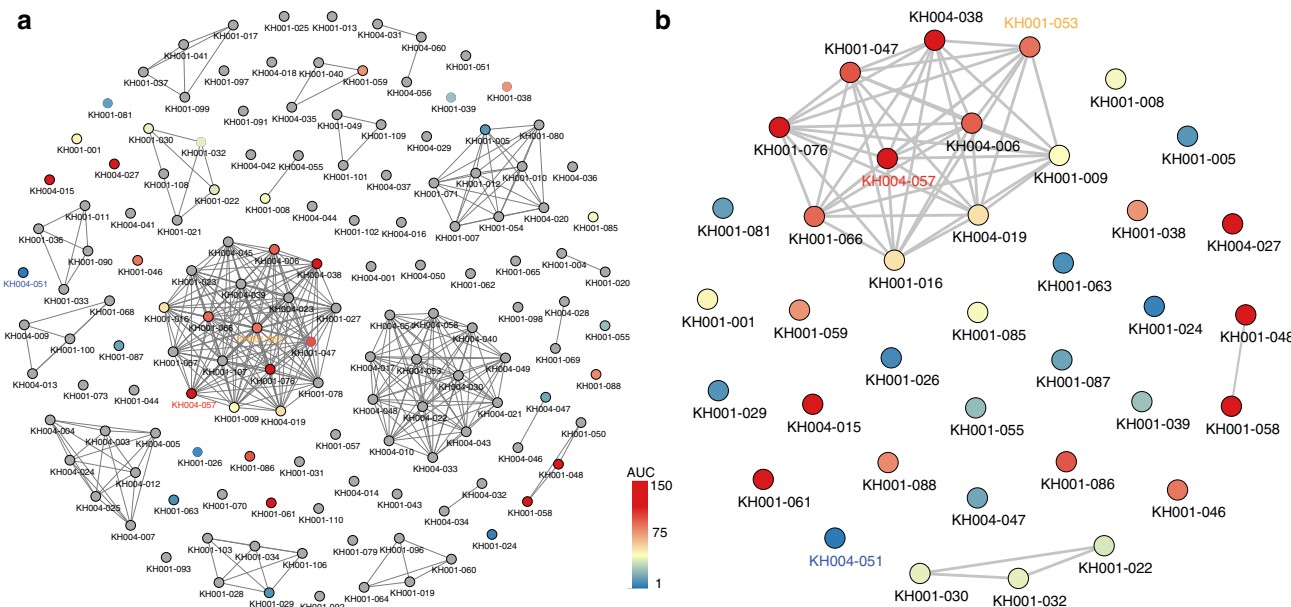

**Fig. 2** PPQR parasites detected within both highly related and unrelated parasite populations. Identity By Descent (IBD) analysis was performed on 157 parasite isolates and a gray line connects parasites sharing > 90% of their genome sequences **a**. Parasites that were phenotyped for PPQ resistance are shown in color, and those not tested are shown in gray. PPQR and PPQS parasites are shown in red and blue, respectively, with the scale of AUC shown. A representative PPQS parasite (KH004_051) is labeled blue, an intermediate PPQR parasite (KH001_053) orange, and a highly-PPQR parasite (KH004_057) red. **a** shows all 157 parasites genotyped, and **b** shows all the phenotyped parasites from **a**

These data indicate that the presence of ART resistance is not required for PPQ resistance, but that parasites with the highest levels of PPQ resistance were also ART resistant. Furthermore, these data suggest that PPQ resistance occurs on multiple genetic backgrounds in this population, rather than in a single parasite lineage.

**AUC is associated with previously identified CNVs and SNPs.** We tested for CNVs for which copy number was either positively (*plasmepsin II–III*) or negatively (*pfmdr1*) associated with PPQ resistance[18,19]. For this analysis we first used whole-genome sequencing (WGS) data for these isolates obtained from the Pf3k project[29] to estimate *pfmdr1* and *plasmepsin II–III* CNV (Supplementary Data 1), and then applied quantitative real-time PCR (qPCR) to corroborate these data among a subset of samples to confirm both DNA copy number (Spearman: *pfmdr1*: $r = 0.6$, $p = 0.003$, $N = 20$, *plasmepsin II*: $r = 0.7$, $p = 0.0008$, $N = 20$) and to correlate RNA expression levels with those CNV levels. In accordance with previous results[18,19], we detected a significant positive correlation between AUC and *plasmepsin II–III* copy number (Spearman: *plasmepsin II*: $r = 0.53$, $p = 0.0007$; *plasmepsin III*: $r = 0.54$, $p = 0.0006$, Fig. 3, Supplementary Fig. 4) and a negative correlation between AUC and *pfmdr1* copy number (Spearman: $r = -0.55$, $p = 0.0004$, Fig. 3). We found a positive correlation between *pfmdr1* copy number and both MEF and lumefantrine (LUM) EC$_{50}$ values (Spearman: $r = 0.61$ and $r = 0.70$, respectively; $p < 0.0001$); and, a negative correlation between *pfmdr1* copy number and chloroquine (CQ) EC$_{50}$ (Spearman: $r = -0.34$, $p = 0.043$, Supplementary Fig. 5). There was no correlation between *pfmdr1* copy number and ART EC$_{50}$. We also detected a significant negative correlation between *plasmepsin II–III* copy number and both MEF and LUM EC$_{50}$ values (Spearman: $r = -0.39$, $p = 0.0169$; and $r = -0.38$, $p = 0.0202$; respectively), but not for CQ or ART EC$_{50}$s (Supplementary Fig. 5). Although a general association was observed between AUC and either increased *plasmepsin II–III* or decreased *pfmdr1* copy number, there were discordant isolates. Specifically, KH001_026 and KH001_081 were highly sensitive to PPQ, but both had increased *plasmepsin II–III* and *pfmdr1* copy numbers (Fig. 3). On the other hand, KH001_061 was highly resistant to PPQ, yet harbored a single copy of both *plasmepsin II–III* and *pfmdr1* (Fig. 3). We tested *plasmepsin II* and *pfmdr1* expression levels in two of the discordant and two control parasites to confirm that expression of *plasmepsin II* and *pfmdr1* was higher in isolates with increased copy numbers (Supplementary Fig. 6).

In addition to the CNVs, several single nucleotide polymorphisms (SNPs) have been associated with PPQ resistance[18,19]. A candidate locus association study was carried out to test whether AUC tracks with any of these previously identified SNPs in this independent set of Cambodian parasite samples (Supplementary Fig. 7). Twelve of 17 loci identified by Amato et al.[18] (Supplementary Data 2) exhibited a significant association

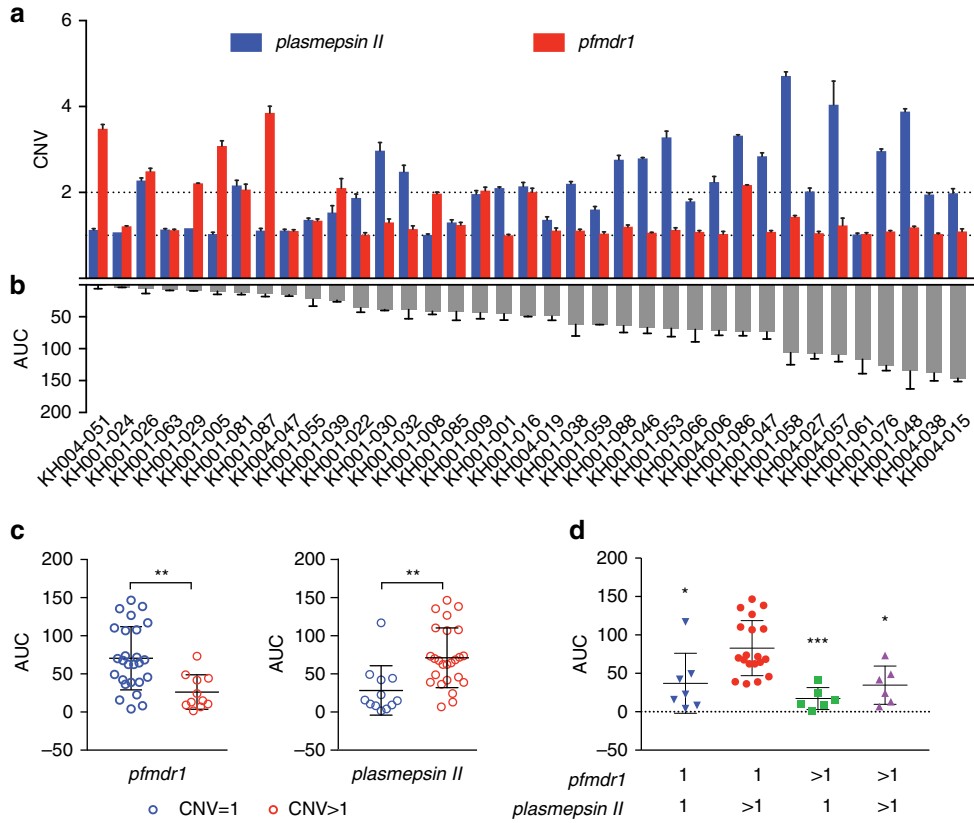

**Fig. 3** AUC correlates overall with *plasmepsin II* copy numbers but there are exceptions. **a** shows CNVs for *plasmepsin II* (blue) and *pfmdr1* (red) that were estimated based upon average read depth of whole-genome sequencing data (+ s.d.) and confirmed by quantitative real-time polymerase chain reaction. **b** shows the Area Under the Curve (AUC) for the parasite lines tested ($N = 3$, mean+ s.d.). **c** shows the distribution of all parasite lines binned into CNV of 1 (blue) or> 1 (red, average > 1.6) for *pfmdr1* or *plasmepsin II*, relative to the AUC values, to show the negative correlation for *pfmdr1* (unpaired Student's t-test, $p = 0.0021$, ** $p < 0.01$) and the positive correlation for *plasmepsin II* ($p = 0.0023$). **d** Combining *pfmdr1* and *plasmepsin II* CNVs, only parasites with *pfmdr1* = 1 and *plasmepsin II* > 1 CNVs are significantly different than the other combinations in regards to the AUC (one-way ANOVA, Tukey corrected). * $p < 0.05$, ** $p < 0.01$. Each data point in **c** and **d** represents the mean of three independent assays run in triplicate. The overall mean and s.d. is shown with black bars

(unpaired Student's t-test, $p < 0.05$) with AUC, strengthening their possible role in modulating parasite responses to PPQ. Included in these loci is the exonuclease (PF3D7_1362500) identified as top hit by Amato et al.[18]. In contrast, only one of 19 SNPs reported by Agrawal et al.[22] was associated with PPQ resistance in this population. Several *pfcrt* mutations have been associated with PPQ resistance in vitro, but most of the isolates in this study had the same Dd2-like *pfcrt* genotype, with only two isolates harboring additional mutations (KH004-027: G353V, KH004-015: M104K, Supplementary Data 3). Thus, we were unable to detect any association between AUC and *pfcrt* mutations among these isolates.

**Plasmepsin II–III CNVs contribute to PPQ resistance.** To further characterize the role of *plasmepsin II–III* CNVs in PPQ resistance, we cloned the KH001_053 parasite isolate by limiting dilution and derived nine sub-cloned lines. All subclones and the initial patient isolate had the same 24-SNP molecular barcode genotype[32] and exhibited a bimodal response to PPQ (Fig. 4a). WGS was performed on three of these subclones to confirm that identical molecular barcode genotypes represented isogenic lines —parasites that have the same or closely similar genotypes. These data confirmed that the bimodal response detected in the PPQ dose–response testing (Fig. 1) was not due to a mixture of parasite genotypes within the culture-adapted sample. All individual clones had only a single copy of *pfmdr1*, but variable copy numbers of *plasmepsin II–III* (ranging between 1 and 2.7 by qPCR). This provided an opportunity to test the impact of *plasmepsin II–III* CNV in the context of a single *pfmdr1* locus among otherwise genetically identical parasites (based upon the molecular barcode or WGS data). Evaluation of the PPQ response among these isogenic clones revealed a range of AUC values that tracked with *plasmepsin II–III* copy number (Fig. 4a). That these isogenic lines all harbored a single copy of *pfmdr1*, confirmed that *plasmepsin II–III* levels play a key role in PPQ resistance in the

absence of changes in *pfmdr1* copy number. To further corroborate these findings, we analyzed 10 parasite isolates with PPQ response data from among the highly related cluster (Fig. 4b). Although these isolates share > 90% IBD by SNP variants across the genome they carry different copy numbers of *plasmepsin II–III* and *pfmdr1* and range in their AUC response to PPQ. All isolates except KH004_019 had more than one *plasmepsin II–III* copy (Fig. 4b) and two isolates had more than one *pfmdr1* copy (KH001_009, KH001_016). These three parasite isolates also had the lowest AUC, further confirming that their increased *plasmepsin II–III* CNV has an important role in PPQ resistance in the context of a single *pfmdr1* copy.

**Stage-specific enhanced survival under high PPQ pressure.** To better understand the biology of the observed bimodal response to PPQ for resistant parasites, we evaluated the effect of PPQ exposure on different lifecycle stages, comparing a PPQS isolate (KH004_051) and a PPQR isolate (KH004_057) at 12-hour intervals to determine whether PPQ exhibits a stage-specific mode of action. Drug-treated and non-drug-treated control parasites were examined throughout the 48-hour life cycle at 12-hour intervals by microscopy (Fig. 5a) and parasitemia was determined by fluorescence-activated cell sorting (FACS) ~ 72 h from the start of the experiment (Fig. 5b). Using parasitemia as a proxy for parasite viability, we analyzed the PPQS and PPQR isolates exposed to five different PPQ concentrations ranging across the concentrations used to calculate the AUC (40 nM, 200 nM, 580 nM, 2 μM, and 10 μM). A 12-hour exposure to 40 nM PPQ had no effect on either the PPQS or PPQR parasite (Fig. 5b). However, exposure of the PPQS isolate to all other concentrations of PPQ for 12 h at any stage prevented completion of the growth cycle, except when drug was applied at the latest stages (36–48 h) where a few parasites could reinvade and form new ring-stage infections (Fig. 2b). Growth arrest was evident by either condensed pyknotic parasite forms or parasites with enlarged

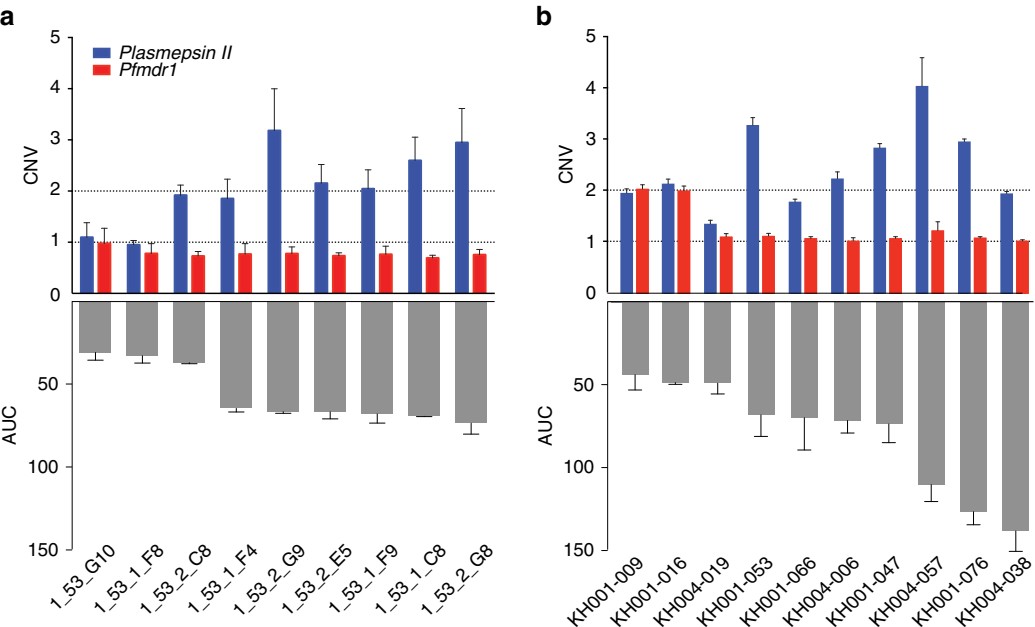

**Fig. 4** Isogenic lines confirm association between *plasmepsin II* CNV and AUC. **a** Subclones of KH001_053, identical by a 24-SNP barcode (all clones) and WGS (3 sequenced clones), had variable *plasmepsin II* CNV levels but only a single *pfmdr1* copy. **b** Isolates with > 90% relatedness vary in *plasmepsin II* and *pfmdr1* CNVs. *Plasmepsin II* CNV levels correlated with AUC values, confirming that this locus has a role in conferring PPQ resistance. All drug assays were done in triplicate and repeated two times, shown are the means with s.d. CNVs are shown as mean+ s.d. from three biological replicates done in quadruplicate by qPCR **a** or as average read depth of whole-genome sequencing+ s.d. in **b**

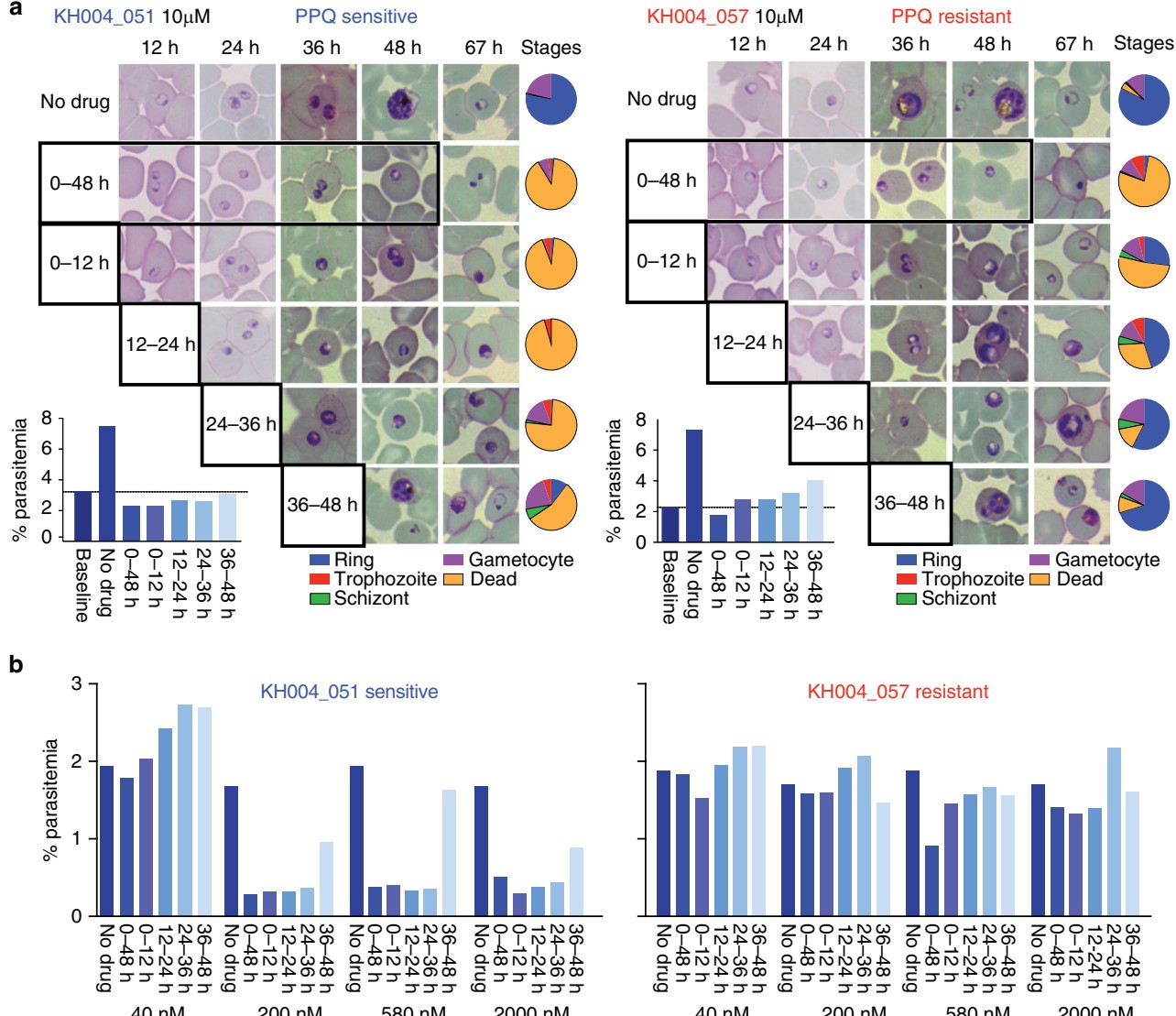

**Fig. 5** Late-stage parasites are less susceptible to PPQ and resistant parasites can tolerate high PPQ doses for 12 h. PPQS (KH004_051, blue) and PPQR (KH004_057, red) were synchronized and exposed to PPQ for 12 h, and their ability to survive to the next life cycle was visualized by microscopy **a** to determine both stage and parasitemia **b**. Representative images **a** and parasite stage distribution (blue = ring, red = trophozoite, green = schizont, purple = gametocyte, and yellow = dead) were calculated and represented by pie charts. Parasitemias resulting from these PPQ exposures were analyzed by FACS 72 h after exposure **b**. Under high drug concentrations (10 μM, **a**), there is evidence that the PPQR parasite (KH004_057, red) can tolerate 12 h of PPQ exposure up to 2000 nM. Conversely, the PPQS parasite (KH004_051, blue) does not survive at PPQ concentrations of ≥ 200 nM, except a few parasites that survive in the later (36–48 h) time of exposure, where there was evidence of some parasite survival

vacuoles (Fig. 5a), and by reduction in parasitemia compared to controls (Fig. 5b). Gametocytes seemed unaffected by drug exposure (Fig. 5a). In contrast, exposure of the PPQR isolate to similar drug concentrations revealed partial completion of the lifecycle when drug was applied for 12 h, evident by the presence of ring-stage parasites and increased parasitemia over baseline in the subsequent cycle. These data indicate that PPQ acts primarily on parasite stages younger than 36 h, and that a 12-hour exposure to ≥ 200 nM PPQ is sufficient to kill PPQS parasites. In contrast, PPQR parasites could complete their lifecycle and reinvaded uninfected erythrocytes after a 12-hour exposure of up to 10 μM PPQ. A modified PSA (mPSA) method using a higher concentration of PPQ (20 μM) was used to confirm that PPQR parasites had consistently higher survival rates at 20 μM PPQ than parasites exposed to 200 nM PPQ, indicating that PPQR parasites grow better under higher PPQ concentrations (paired Student's *t*-test, *p* = 0.0021, Supplementary Fig. 8).

## Discussion

The emergence of PPQ resistance among *P. falciparum* in Cambodia raises concerns about both the utility of ACTs like DHA–PPQ for treatment in SEA, and the potential risk for these drug resistant parasites to spread to other geographical settings where DHA–PPQ is being used. To explore mechanisms of PPQ resistance we leveraged cultured parasites from Cambodia, where clinical PPQ resistance has recently emerged, to identify phenotypic and genotypic characteristics of PPQ response among these parasites that may help elucidate possible mechanisms of clinical PPQ resistance observed among patients. Current PPQ response assays are often unreliable or time consuming, so we developed a new, robust method to quantify PPQ resistance. Our investigation of recent patient isolates from Cambodia revealed that parasites exhibit a bimodal response to increasing PPQ concentrations that can be quantified by an AUC value. Using isogenic parasite lines we demonstrated a primary role for increased *plasmepsin II–III*

copy number for PPQ resistance in the presence of a single copy of *pfmdr1*. Biologically, PPQR parasites survive high PPQ concentrations and can withstand 12-hour PPQ exposure up to concentrations of 10 μM. This bimodal phenotype for PPQ suggests a biological survival mechanism for PPQR parasites under high drug concentrations.

PPQR parasites do not show a typical shift in $EC_{50}$ values in conventional drug assays, but rather yield non-interpretable dose–response curves[10,33–35] that are often excluded from analysis. By increasing both the PPQ starting concentration and the number of concentrations tested, we identified a new and unusual PPQ phenotype whereby parasites survive better under higher PPQ concentrations than under some lower concentrations of drug. Other groups have likely missed this phenotype when using $EC_{90}$ values or excluding data points as outliers[22,35,36]. A collection of evidence, including use of [$^3$H]-hypoxanthine incorporation and visualization of viable parasites, indicates that parasites exhibiting a bimodal response with high AUC values can survive these elevated PPQ concentrations (2 μM) better than low concentrations (200 nM). This is reminiscent of the survival of certain bacterial strains to penicillin and other beta lactam antibacterials at higher concentrations, and has been termed the "Eagle effect"[37,38]. AUC data correspond with previously reported PSA phenotypes and correlate with *plasmepsin II–III* copy number, demonstrating that the bimodal response with high AUC is a valid phenotype for PPQ resistance. The AUC assay is advantageous because it is easy, quick, and robust and does not require tight synchronization in vitro, or counting parasitemia by microscopy. AUC provides a broad dynamic range for PPQ response and may reveal intermediate phenotypes (AUC = 35–100) that may be useful for monitoring the emergence and spread of PPQ resistance as many of these intermediate isolates had increased *plasmepsin II–III* copy numbers. This bimodal PPQR phenotype suggests a unique mechanism of survival under high PPQ concentration. For example, perhaps PPQ induction of stress response pathways enables PPQR parasites to survive under high PPQ levels. Studies of this response can guide our understanding of PPQ resistance mechanisms to help explain how PPQR parasites survive drug exposure.

The association of PPQ resistance with increased *plasmepsin II–III* and decreased *pfmdr1* copy numbers has been observed before; however, without functional analysis it was unclear how these two detected changes contribute to PPQ resistance. Using our highly related parasite lines we demonstrated the positive effect of increased *plasmepsin II–III* copy numbers on PPQ resistance on the genetic background of a single *pfmdr1* copy. These data strengthen the role for *plasmepsin II–III* in PPQ resistance in the absence of any compensatory changes in *pfmdr1* copy number, but do not rule out a separate role for *pfmdr1* amplification in decreasing PPQ susceptibility. Interestingly, these parasites are highly related to each other by SNP genotyping but vary in their *plasmepsin II–III* copy numbers suggesting several recent duplication events. Finding different *plasmepsin II–III* copy numbers in a single sub-cloned isolate could indicate a fitness cost for the duplication and that loss is frequent, or that parasites lose multiple copies upon culture adaptation in vitro. That we did not see an overall loss of CNV in two isolates that were cultured in vitro for over 6 months suggests that the pressure in vivo might be more important to fitness. However, there is an overall decrease of *pfmdr1* copy numbers observed in Cambodia, which has encouraged reintroduction of AS-MEF for treatment in this region[39]. In addition, a triple ACT strategy (ART+ MEF+ PPQ) is under evaluation in TRAC II[40]. On the other hand, emergence of PPQ resistance could jeopardize the proposed use of the drug in combination with new partners for

non-ACTs for malaria such as fosmidomycin–PPQ, which is under evaluation in Phase II clinical trials[41].

As to the mechanism of PPQ resistance involving *plasmepsin II–III*, PPQ is thought, like CQ, to accumulate in the food vacuole (FV) and inhibit conversion of toxic heme moieties to non-toxic hemozoin crystals during hemoglobin digestion. We have shown that PPQ is acting on all stages of the parasite life cycle but is least effective at the latest stages (Fig. 5). High-resolution live-cell imaging has shown that hemoglobin degradation is initiated early after invasion of the RBC by the parasite[42]. These authors suggest that cytostome-mediated endocytic events are the first step in the genesis of the hemoglobin-degrading apparatus of *P. falciparum*. Endocytosis of the host compartment is then followed by a process that concentrates the contents of the endocytic compartment and hemozoin crystals start to accumulate in late ring-stage parasites. It is therefore not surprising that PPQ is acting on early stages as well as later stages. There is less data available on hemoglobin degradation in late trophozoites/schizonts but it is possible that degradation slows down once parasites are converted into merozoites and are ready to egress. *Plasmepsin II*, *plasmepsin III*, and *pfmdr1* are located in the food vacuole, with *plasmepsin II* and *III* directly involved in hemoglobin degradation generating small peptides and eventually amino acids for protein synthesis but also the toxic byproduct heme. Treatment of parasites with PPQ leads to a depletion of ribosomes and swelling of the FV with undigested hemoglobin vesicles and reduced hemozoin crystals[43] as well as an accumulation of free heme[21]. Although the free heme is toxic for the parasite, inhibition of hemoglobin degradation will starve the parasite and increased *plasmepsin II–III* might help the parasite to maintain amino-acid production when hemoglobin degradation is otherwise inhibited by PPQ. More experiments are needed to understand how plasmepsin *II–III* and PPQ influence hemoglobin degradation and the role of this process in PPQ resistance.

In addition to demonstrating a primary role of *plasmepsin II–III* CNV in PPQ resistance in the absence of *pfmdr1* CNV in this population, we found a significant correlation between a majority of previously identified SNPs and PPQ resistance by AUC phenotype. Interestingly, in this population only two haplotypes were seen surrounding a putative exonuclease SNP that was most strongly associated with PPQ resistance previously[18]. One of these haplotypes is completely absent from the PPQS population. Although a majority of the PPQ phenotypes can be explained by *plasmepsin II–III* CNV, there were four discordant parasite lines in this study. None of the proposed SNPs could explain the discrepancy between the PPQ phenotype and *plasmepsin II–III* copy number. Mutations in *pfcrt* that are associated with PPQR were not detected among these parasites. Population structure analysis indicates highly related parasite clusters, but no evidence of simple clonal expansion of a single PPQR parasite in this population. *Pfkelch13* genetic variation did not correlate with PPQ resistance, and both PPQS and PPQR parasites had evidence of the common C580Y genetic background. The only *PfKelch13* mutation exclusively identified in PPQR parasites was Y493H, found among highly related and unrelated subpopulations. These data suggest that PPQ resistance developed several times on at least two different *Pfkelch13* backgrounds.

Understanding mechanisms of PPQ resistance is paramount to successful control and elimination of *P. falciparum*. The bimodal PPQ phenotype suggests these parasites may induce biological responses that promote survival under high PPQ concentrations. Coupled with the role of *plasmepsin II–III* in modulating PPQ response, additional genetic or expression variation may be important for this antimalarial phenotype.

Definition of the AUC phenotype and the parasite lines derived in this study provide key resources to aid our understanding of biological changes required to attain high levels of PPQ resistance and can promote strategies to track resistance and identify alternative antimalarial strategies to circumvent or overcome this resistance phenotype in *P. falciparum*.

## Methods

**Culture adaptation and maintenance of TRAC parasites**. All parasite samples were collected under protocols approved by ethical review boards in Cambodia, at Oxford University (OxTREC), National Institutes of Allergy and Infectious Diseases, and at the Harvard T.H. Chan School of Public Health. Culture adaptation of parasites was accomplished by thawing cryopreserved material containing infected red blood cells (iRBCs) that had been mixed with glycerolyte. Parasites were maintained in fresh human erythrocytes (O+) in Hepes buffered Rosewell Park Memorial Institute (RPMI) media containing 10% O+ human serum (heat inactivated and pooled). Human blood products (erythrocytes and serum) were obtained from Interstate Blood Bank, Inc., Memphis, TN. Cultures were placed in modular incubators and gassed with $1\%O_2/5\%$ $CO_2$/balance $N_2$ gas and incubated with rotation (50 rpm) in a 37 °C incubator. Parasite lines were frozen within a few cycles (< 2 weeks) after parasites first appeared; and, never maintained for longer than 2 months in culture after these initial stocks were thawed. The 37 parasites used in this study were selected from among 157 previously adapted lines[25]. An initial sample subset ($n = 28$) was randomly identified, but an additional set ($n = 9$) was included based upon their barcode identity to KH001_053 to enrich for parasites that might have genetic variants that contribute to PPQ resistance.

**In vitro 72 H drug susceptibility by SYBR green staining**. Drug susceptibility was measured using the SYBR Green I method as previously described[44]. In brief, tightly synchronized 0–6 h rings were grown for 72 h in the presence of different concentrations of drugs in 384-clear-bottom well plates at 1% hematocrit, 1% starting parasitemia and 40 μl of 0.5% Albumax culture media. Growth at 72 h was measured by SYBR Green I (Lonza, Visp, Switzerland) staining of parasite DNA. A 24-point dilution of PPQ (Sigma-Aldrich, St. Louis, MO) and a 12-point dilution series of the rest of the drugs (DHA, ART, AS, MEF, and LUM; Sigma-Aldrich, St. Louis, MO) were carried out in triplicate and repeated with at least three biological replicates. Drug stocks were resuspended in dimethyl sulfoxide except for CQ prepared in 0.1% Triton X-100 in water and PPQ prepared in 0.1% Triton X-100 and 0.5% lactic acid in water to ensure complete dissolution, as lactic acid enhanced PPQ solubility[45]. Drugs dispensed by a HP D300 Digital Dispenser (Hewlett Packard Palo Alto, CA), with the Triton X-100 enhancing dispensing of aqueous drug stocks. Relative fluorescence units was measured at an excitation of 494 nm and emission of 530 nm on a SpectraMax M5 (Molecular Devices Sunnyvale, CA) and analyzed using GraphPad Prism version 7 (GraphPad Software La Jolla, CA). $EC_{50}$ values were determined with the curve-fitting algorithm log (inhibitor) vs. response–Variable slope except for PPQ. PPQs bimodal dose–response would not allow for any curve-fitting hence susceptibility was measured by calculating the area under the second response curve (AUC). AUC was calculated by fitting a six-dimensional least-square polynomial equation to each sample's dose–response curve and integrating the fitted equation over the region corresponding to the second response curve. These equations were fit to the data using the polyfit module in NumPy, a fundamental package for scientific computing using Python. We chose to use a six-dimensional least-square polynomial equation because it captures the dynamics of the second response curve better than equations with fewer dimensions. Using equations with dimensions > 6 do not result in major differences in calculated AUC values. In fact, AUC values calculated using the boundaries identified by our equation fitting method (0.10 μM – 30 μM) gave roughly the same result. The boundaries of the second response curve were determined by identifying the drug concentration corresponding to the average local minima before and after the second response curve across all drug-assayed samples.

Spearman correlation analysis was performed to assess the relationship between the antimalarial $EC_{50}$ values and PPQ AUC, in vivo clearance half-life, ring survival assay survival rate or *pfmdr1* copy number. *P* values < 0.05 were considered significant.

**[³H]-hypoxanthine incorporation assays**. To determine whether the PPQ bimodal drug dose–response observed in the SYBR Green I method is consistent with another drug susceptibility assay, [³H]-hypoxanthine incorporation of live parasites was measured. This assay is based on the previously published method of Desjardins et al.[46] with some modifications. Tightly synchronized 0–6 h ring-infected erythrocytes were plated at 2% hematocrit and 0.5% parasitemia in 100 μl volumes in a 96-well plate. The RPMI media used to suspend the parasites contained 1/20th the amount of hypoxanthine normally used in parasite in vitro culture (2.8 mg/L) and then supplemented with 5% Albumax and gentamicin. The 96-well plates used contained pre-dispensed PPQ as described above. The parasites were incubated for 72 h until the next ring stage after which 20 μl of [³H]-hypoxanthine (50 μl/ml of hypoxanthine-low media, PerkinElmer Waltham, MA)

was added to each well. The parasites were incubated further for 37 h until the trophozoite stage, lysed by freezing at −80 °C overnight, parasite DNA transferred from lysed cells to 96-well filter plates (UniFilter GF/B PerkinElmer Waltham, MA) using a cell harvester (Filtermate Harvester, Packard PerkinElmer Waltham, MA), 30 μl each of microscintillation fluid (Microscint-O, PerkinElmer Waltham, MA) added per well and radioactivity measured in a microplate scintillation counter (TopCount-NXT, Packard PerkinElmer Waltham, MA).

**Copy number variation assay**. To determine copy numbers of *pfmdr*1, *plasmepsin* II, and III, qPCR was performed on genomic DNA (extracted with QIAmp Blood Mini Kit, Qiagen, Hilden, Germany) as previously described[47] with the following modifications: amplification reactions were done in MicroAmp 384-well plates in 10 μl SYBR Green master mix (Applied Biosystems, Foster City, CA), 150 nM of each forward and reverse primer and 0.4 ng template. Forty cycles were performed in the Applied Biosystems ViiA™ 7 Real-time PCR system (Life Technologies, Carlsbad, CA). *pfmdr1* forward (5'-TGCATCTATAAAACGAT CAGACAAA-3') and reverse primers (5'-TCGTGTGTTCCATGTGACTGT-3') were designed after Price, et al.[48], whereas *β-tubulin* forward (5'-CGTGCTGGCC CCTTTG-3') and reverse (5'-TCCTGCACCTGTTTGACCAA-3') primers for the endogenous control were designed after Ribacke, et al.[47]. Two primer sets were designed for *plasmepsin II*: 1st set: forward (5'-TCCTTGGTTTAGGATGGAA AGA-3') and reverse 5'-CCACCAATGGTTAAGAATCCTG-3') 2nd set: forward (5'-CCATTGGTGGTATTGAAGAAAGA-3') and reverse (5'-TTTCCAACGT GTGCATCTAAA-3'); and, one set for *plasmepsin III*: forward (5'-GGTAGTGA GTTTGATAATGTGG-3') and reverse (5'-CACAAGACTCTGATGTACA-3'). Technical replicates were run in quadruplicates. Copy numbers were considered increased (> 1) when the average of three biological replicates was above 1.6.

**PSA and modified PSA**. The PSA was performed as described previously[16]. In brief, 0.75% 0–3 h ring-infected erythrocytes in 2 ml 0.5% Albumax culture media at 2% hematocrit were cultivated without drug and with 200 nM PPQ in 24-well plates. After 48 h, cultures were washed, resuspended in 2 ml 0.5% Albumax culture media and further incubated for 24 h. Thin blood smears were prepared and Giemsa stained. The proportions of viable parasites in exposed and unexposed cultures or the PSA survival rate (%) were evaluated by microscopic assessment of at least 20,000 erythrocytes infected with second-generation rings and trophozoites with normal morphology by two blinded microscopists; and, by using FACS with SYBR Green I as described below. Only those samples with ≥ 1.5 × growth rates (parasitemia at 72 h/parasitemia at 0 h) were deemed interpretable for PSA survival rate.

In addition, a mPSA assay was performed as described above with the following modifications: In brief, parasites were exposed to the drug for 72 h. Moreover, 2 μM PPQ concentration was tested in addition to 200 nM, and microscopic assessment was done for ≥ 500 erythrocytes infected with second-generation rings and trophozoites with normal morphology. Those with ≥ 2% parasitemia for the unexposed cultures were deemed interpretable for mPSA survival rate.

**FACS with SYBR Green**. Parasites were stained in 10 × SYBR Green I in 1 × PBS for 30 min in the dark at 37 °C. The staining solution was removed and cells were resuspended in five times the volume of the initial volume of PBS. FACS data acquisition was performed on a MACSQuant VYB (Milteni Biotec) with a 488 nm laser and a 525 nm filter and analyzed with FlowJo 2. RBCs were gated on the forward light scatter and side scatter and infected RBCs were detected in channel B1. At least 100,000 events were analyzed per sample.

**Parasite set up for drug exposure at different time points**. For PPQ exposure experiments at different time points, a PPQS (KH004_051) and PPQR (KH004_057) parasite were used. In brief, tightly synchronized 0–3 h ring-infected erythrocytes were plated at 1 or 2% parasitemia, 2 or 2.5% hematocrit and in 2 or 4 ml volumes of 0.5% Albumax culture media in a 24- or 12-well plate. Parasites were incubated without PPQ and in 40 nM, 200 nM, 580 nM, 2 μM or 10 μM PPQ at the following time points: 0–12 h, 0–24 h, 12–24 h, 24–36 h and 36–48 h. Parasitemia was assessed by microscopic examination of stained blood smears and/or by FACS of SYBR Green-labeled samples taken every 12 h until the next generation of rings at 67–72 h.

**Genotyping**. Cultured parasite lines were genotyped using a molecular barcode[32] to confirm that the cultured line matched the original patient material and harbored only a single parasite genotype.

**WGS analysis for SNPs and CNVs**. This publication uses data generated by the Pf3k project (www.malariagen.net/pf3k)[29]. Except for the KH004-057 and the KH001-053 subclones (data deposited: https://www.ncbi.nlm.nih.gov/bioproject/414203), we used the variant call format (VCF) files provided by the Pf3k project to perform whole genome analyses. For KH004_057 and the KH001_053 subclones, gDNA was extracted and sheared with a Covaris S220 Focused-ultrasonicator (Covaris, Woburn, MA, USA). Illumina-compatible libraries were prepared on the Apollo 324 (WaferGen Biosystems, Fremont, CA, USA) and sequenced on an

Illumina HiSeq 2000 (Illumina, San Diego, CA, USA). *P. falciparum* populations were sequenced with the goal of reaching over 60 × average fold-coverage across the genome. Reads were aligned to the *P. falciparum* 3D7 reference assembly (PlasmoDb v 7.1) using the Burrows-Wheeler Aligner (version 0.5.9-r16)[49]. Variant calls were determined using the GATK Unified Genotyper[50] using the parameter and quality thresholds described in the supplementary information of a previous paper[30]. The resulting VCF files were then combined with the VCF files downloaded from the Pf3k project and filtered to remove non-variant sites using VCFtools[51].

To determine whether any previously implicated drug resistance SNPs were associated with PPQR, we analyzed the SNPs in the genes listed in the Supplementary Data 2. Samples were divided into two categories based on their AUC values. Samples with an AUC ≤ 35 were considered PPQS and samples with AUC > 35 were considered PPQR. Based on these categories, we used the software package PLINK (version 1.8)[52] to determine whether any SNP was associated with PPQR.

To assess CNV, we quantified the fold-change in the average read depths of non-variant sites within *plasmepsin I* (PF3D7_1407900), *plasmepsin II* (PF3D7_140800), and a nearby conserved gene of unknown function (PF3D7_1408200) as compared to the average read depths of non-variant sites in several randomly chosen regions of the genome (nucleotide positions at Chr 3: 353552–361453, Chr 5: 322315–329693, Chr 14: 578376–603856). These regions contain mostly coding sequence and have little variation in read depth. Genes within these regions are annotated as: a putative DNA polymerase epsilon subunit (PF3D7_0308000), a 6-cysteine protein (P38) (PF3D7_0508000), a putative atypical protein kinase, ABC-1 family (PF3D7_1414500), an RNA guanyltransferase (PF3D7_1414600), a putative carboxyl-terminal hydrolase (PF3D7_1414700), a putative small nuclear ribonucleoprotein-associated protein B (PF3D7_1414800), and several conserved proteins of unknown function (PF3D7_0308100, PF3D7_0507800, PF3D7_0507900). P38 is suspected to be involved with gamete fertilization[53] and under some level of immune-mediate balancing selection[54], but little is known about the functions of the other genes. None have been implicated in drug resistance studies.

**Identity by descent analysis**. The relatedness between two strains was calculated as the proportion of the genome inherited from the same ancestor, or identical by descent. Relatedness was calculated from SNP data using a hidden Markov model described in[31]. Networks of relatedness were generated using NetworkX (https://github.com/networkx/networkx), a Python package for the creation, manipulation, and study of complex networks, and visualized using Gephi[55], an open-source software for network visualization and analysis.

**Statistical analysis**. Calculations of $EC_{50}$, Student's *t*-test, one-way analysis of variance and correlations were performed using GraphPad Prism version 7 (GraphPad Software La Jolla, CA).

**Data availability**. WGS data that support the findings of this study have been deposited in NCBI bioproject with the PRJEB22985 accession codes (https://www.ncbi.nlm.nih.gov/bioproject/414203) or was provided by Pf3k project (www.malariagen.net/pf3k)[29]. TRAC study samples are made available to the malaria research community through the worldwide antimalarial research network (WWARN): http://www.wwarn.org/news/news-articles/trac-study-samples-be-made-available-malaria-research-community. All other data supporting the findings of this study are available within the article and its Supplementary Information files, or are available from the authors upon request.

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

## Acknowledgements

We thank Courtney Edison for slide reading and subcloning the KH001_053 isolate. The computations in this paper were run on the Odyssey cluster supported by the FAS Division of Science, Research Computing Group at Harvard University. N.J.W., C.J.W. were supported by Bill and Melinda Gates Foundation Global Health Grant Number OPP1040463, which also funded culture adaptation of the parasites described herein. Additional support was provided to DFW by the Bill and Melinda Gates Foundation Global Health Grant Number OPP1053604. This document is an output from a project of the Tracking Resistance to Artemisinin Collaboration funded by the UK Department for International Development (DFID) for the benefit of developing countries. However, the views expressed and information contained in it are not necessarily those of or endorsed by DFID, which can accept no responsibility for such views or for any reliance place on them. This work was supported in part by the Intramural Research Program of the National Institute of Allergy and Infectious Diseases, National Institutes of Health.

## Author contributions

S.B. designed experiments, carried out PCR-based genotyping and copy number variation analysis, and wrote the manuscript; P.M. carried out drug testing, PSA, and copy number variation analysis; and A.M. carried out *Pfkelch13* genotyping. W.W. carried out the genomic analysis with sequencing data; S.F.S. helped with genomic analysis and provided critical edits to the manuscript; P.L., M.D., C.J.W., E.A.A., A.M.D., N.J.W., A.M.D., and R.F. carried out field collection, provided samples and in vivo sample information. D.M. and F.A. shared information about *plasmepsin II–III* before publication; D.F.W. provided critical review of the data and provided experimental guidance. S.K.V. culture-adapted the parasites, carried out molecular barcode genotyping and analysis, provided experimental guidance and critical review of the data, and wrote the manuscript. All authors provided critical review of the data and manuscript before publication. All authors read and approved the final manuscript.

## Additional information

**Competing interests:** The authors declare no competing interests.

