## [Peer Review File · Nature Communications]

Editorial Note: Images have been redacted from this peer review file as indicated to protect copyright claims.

Reviewers' comments:

Reviewer #1 (Remarks to the Author):

This manuscript reports on sensitivity to piperazine in Cambodian *P. falciparum* isolates collected in 2011. The authors identified an unusual bimodal dose-response curve in resistant parasites and showed that resistance was associated with increased plasmepsin copy number. These findings are not novel, as piperazine resistant parasites with this phenotype and genotype have recently been described from Cambodia by multiple groups. The main advance provided in the manuscript is a new assay, which is actually a minor, but well-conceived adjustment of a standard dose-response curve, adding some higher concentrations of piperazine to fully capture the complex pattern of drug sensitivity in the Cambodian parasites. With this larger dose-response curve, the bimodal shape is captured, and an area under the curve (AUC) for a relevant portion of the curve (at higher concentrations of piperazine) can be determined. This AUC is used to define piperazine resistance, offering arguably a better means of doing this than a recently described piperazine sensitivity assay that is in some regards simpler, but requires microscopic counting of parasites; the two assays are also shown to correlate quite well. This manuscript also offers an in depth characterization of genetic relatedness and sharing of drug resistance determinants among the studied Cambodian isolates and characterization of the stage specificity of piperazine. Overall, the manuscript is well-written and clear and the results are well-displayed and communicated. The manuscript offers a more in-depth characterization of the already well-described piperazine resistance genotype/phenotype in Cambodia. Some specific minor concerns are as follows.

- 1) Abstract. The penultimate sentence is confusing. The disconnect between *pfmdr1* copy number and PPQ phenotype is not surprising, and arguably need not be mentioned in the abstract. Rather, a broader summary statement addressing the observation that plasmepsin copy number appeared to explain piperazine resistance in most, but not all cases, would be more helpful.
- 2) Introduction. First paragraph. ACTs are not used only to forestall resistance development, but also (and primarily) because artemisinin monotherapy has unacceptably high failure rates. At the end of the paragraph all references refer to piperazine resistance; at least one reference to support comments about mefloquine resistance would be helpful. Alternatively, mefloquine might not be mentioned, as this report is, of course, about PPQ.
- 3) Supplementary Figure 1b. The units of AUC should be noted.
- 4) Discussion. First sentence. It is unclear why the authors singled out use of DHA-PPQ for MDA. In fact, the drug is widely used for therapy, and its use in MDA remains mostly investigational. It would be more relevant to express concern about all areas where the drug is used, whether for treatment or MDA or IPT (so, consider omitting the last two words of the sentence).
- 5) Discussion. Paragraph beginning "As to the mechanism of PPQ resistance...". The authors

fail to note that increased plasmepsin copy number, which presumably affords greater plasmepsin activity, would be expected to lead to a greater degree of hemoglobin hydrolysis, thus more free heme, and thus GREATER activity of aminoquinolines such as chloroquine and PPQ, which act by preventing heme detoxification. Thus, as has been noted in many a hallway conversation at scientific meetings, the results, although clearly true, defy our understanding of aminoquinoline action and parasite biology. The last sentence of this paragraph ("Increased plasmepsin II-III copies could help parasites overcome inhibition of hemoglobin digestion by PPQ") is misleading. Chloroquine and PPQ can inhibit hemoglobin digestion at very high concentrations, but their primary mode of antimalarial action is believed to be inhibition of heme detoxification, not inhibition of hemoglobin digestion. This is a complicated area and the authors need not dive into it fully, but their superficial discussion is unhelpful, and if they wish to try to explain the biology behind plasmepsin copy number and PPQ action, a more nuanced discussion is advised.

Reviewer #2 (Remarks to the Author):

See uploaded review

[Reviewer 2 Comments]

[A caveat...I know no nothing about measuring viability in vitro or in conducting these types of dose/response assays...I have assumed the methods are appropriate and accurate but strongly recommend the ms is also reviewed by a specialist in this area]

The paper reports an interesting phenomenon i.e. bi-modality of a dose/kill curve (Figure 1) . I enjoyed my first read but became increasingly concerned at subsequent reading as the phenomenon is so poorly described. In particular, the arguments often appear circular i.e. resistant parasites have bi-modal responses so any with bi-modal response are resistant... this becomes a big problem because the causal reader may think that when they talk about PPQ-resistant isolates that the resistance has been observed in patients i.e. as drug failures. Hopefully this is presentational but there are some clarifications that need to be made before publication i.e. as outlined in the “Major points” below.

Major points

(1) Are the assays done under physiologically-plausibly conditions?? They need to put it into context. Their scale is in uM but many field studies report human concentration in ng/ml or mg/ml.... see example from Saunders et al at the end of this review. They should provide a conversion (most easily in the captions to Table 1) so readers can compare with concentration noted in patient’s post-treatment (ng.ml, mg/ml). They use 72 hour exposure for Fig 1, whereas PPQ is believed to persist at active concentrations for weeks post-treatment (e.g. the Saunders et al graph below) so they must include an explicit, objective discussion of this.

(2) Figure 1.. they need to say that exposure was for 72 hours but, more importantly, that the parasites were “tightly synchronised 0-6 hour rings” (quote from the methods). Why were these ring stages selected over other stages (see also point 3 below)? They later look at different stages (Figure 5) but under different assay conditions (12 hour exposure)... so the question is whether this bimodal phenotype only occurs in rings or whether it occurs in other stages This is major omission and needs to be discussed. I know how labs work and it may well be case that, in retrospect, they should have tested all the stages in the same way but they need to discuss this. I looked at the histogram on fig 5 and can see no clear signal of bimodality in later stages

(3) There is no discussion about the implications of stage-specific killing of the drug. PPQ is believed to interfere with haem metabolism (as noted on page 14) and I think the consensus is that it kills stages around 20 to 45 hours post-invasion when this metabolism occurs. The impact of this should be discussed. For example, figure 5 is based on 12 hour exposures... those parasites synchronised to ages 0 to 12 will only get a few hours killing (i.e. as they develop to age 20+ hours) how will this affect the dynamics??

(4) the most interesting facet of this paper of the bi-modal sensitivity. It is not clear how consistent this is either over stages (see above) or over isolates. The caption to Figure 2 states “ Bimodal growth was observed with increasing PPQ concentration for a subset of parasites”. This needs to be quantified...did the “subset” consist of a single isolate (KH0004_57) or many isolates? The answer is

the latter but we need to go to SI figure 1 to find this information which should be in the main text. Even then it is not easy to visualise individual curves. Did they assign something as being non-bimodal vs bimodal purely by eyeballing each curve?? I would have preferred a more objective criterion e.g “bimodality was defined as the presence of a second peak with 2 or more measures of >xx % growth and separated from the initial curve by at least two points with % growth less than yy” or something similar.

It was also not obvious to me (e.g. when they talk about “discordant” results) whether bi-modality also occurs in the absence of plasmepsin increased copy number and/or whether some isolates with increased plasmepsin copy number show unimodal profiles. This needs to be made explicit in the main text...they also need to be unambiguous when they talk about “discordant” results.

(5) The manuscript lack important details and quantification.

(i) on page 2 they say they tested 37 isolates from 157. On what criteria were these 37 chosen?

(ii) A large proportion of isolates had increased plasmepsin copy number. Was there widespread PPQ failure at the collection sites (NB PPQ+DHA will fail, even if parasites are fully sensitive to DHA...its the partner drug that determines therapeutic outcome) . If there was no evidence of drug failure in patients it suggests the bi-modal phenotype may not yet be clinically relevant. I note they were collected in 2011 so that may be before PPQ resistance spread?

(iii) How long cultured?? The methods has a section called “culture-adaptation” but needs details. Given that subclones of some isolates differ in the copy number it would be nice to know how quickly the adaptation occurred. If it was very rapid then the copy number variation (CNV) may reflect what was in the patient, if very long then CNV may have arisen during the adaptation process.

(iv) I do not like the term “isogenic”. To me (and I suspect many readers) it implies an inbred line genetically engineered to contain only a single genetic difference. In the current case they mean sub-clones and I think they should stick to this term. They might argue that plasmepsin copy number is the only genetic difference between these subclones (and hence they are isogenic) but they have not formally proved it, plus it is hard to understand why plasmepsin CNV should be the only variation within the isolate.

(6). I object to the statements in the first part of the discussion. I don't believe they are supported by the data and need to be toned down. They state “we developed a new, robust method to quantify PPQ resistance” (I think this is tautological but I'll let this pass). The point is they do not demonstrate it has any link to clinical resistance. I think they should be more circumspect. My understanding is that people have now shown the increasing plasmepsin copy number is a risk factor for therapeutic failure so they should say something like “Recent work (citations) has shown a link between increasing plasmepsin copy number and risk of drug failure, our results suggest a mechanism by which these phenotype may arise” or something along those lines. Similarly they state that their results “revealed that PPQR parasites exhibit a bimodal response to increasing PPQ concentration”... this does seem to be tautological... they define PPQR by bimodality, then claim bimodality identifies this phenotype. Overall, they just have to be more circumspect in discussing the implications of their

study. Similarly on page 13 I do not follow/understand their argument about “bona fide phenotypes” and this needs to be clarified

(7) they should re-write the Abstract for the reason given above. For example they say “(AUC) correlating to PPQ resistance. A casual reader would assume this is a correlation to therapeutic outcome. In fact, AUC is their measure of PPQ resistance and the terms Bimodal and PPQR seem to be the same thing to i.e. PPQR is defined as a those isolates with a bimodal curve.

(8) The interesting part of this work is the observation of a bimodal distribution in sensitivity (Fig 1). How do they think this could arise? Could it be an artefact i.e. drug interfering with their assay?? They briefly discuss this on page 13 (“stress response pathways”) but I don’t see how that could generate a bimodal response. Has similar responses been noted to other drugs? If they have no idea why it arises, they shouldn’t be afraid to say this, but readers deserve some discussion as to why it could arise.

Minor points.

(A) Introduction...they imply that artemisinin (partial) resistance has led to the evolution of resistance to other drugs. This is a standard “SE Asia” argument but is controversial... all they need say is that PPQ-DHA is failing, and this is due to PPQ resistance (an effective partner drugs means its ACT will not fail).

Very minor points

(i) Page 4 “PSA” ...has the abbreviation been explained? Figure 3 cited before figure 2. Caption to Fig 3: have “qPCR” and “WGS” abbreviations been given. Page 8 has ‘MEF’ been defined

[redacted]

Taken from Saunders DL, Vanachayangkul P, Lon C. Dihydroartemisinin–Piperaquine Failure in Cambodia. *N Engl J Med* 2014; **371**(5): 484-5. To stop the authors from drawing false conclusions in the traditional game of guess-the-reviewer, I should state that I am not one of the authors of that paper.

We thank the reviewers for their careful review and constructive comments to improve the manuscript and clarify findings from the research. Reviewer comments are in bold for **reviewer 1** and **reviewer 2**, our responses are in *italics*, and references to direct changes to the manuscript are in **red text**.

Reviewer #1 (Remarks to the Author):

This manuscript reports on sensitivity to piperazine in Cambodian *P. falciparum* isolates collected in 2011. The authors identified an unusual bimodal dose-response curve in resistant parasites and showed that resistance was associated with increased plasmepsin copy number. These findings are not novel, as piperazine resistant parasites with this phenotype and genotype have recently been described from Cambodia by multiple groups. The main advance provided in the manuscript is a new assay, which is actually a minor, but well-conceived adjustment of a standard dose-response curve, adding some higher concentrations of piperazine to fully capture the complex pattern of drug sensitivity in the Cambodian parasites. With this larger dose-response curve, the bimodal shape is captured, and an area under the curve (AUC) for a relevant portion of the curve (at higher concentrations of piperazine) can be determined. This AUC is used to define piperazine resistance, offering arguably a better means of doing this than a recently described piperazine sensitivity assay that is in some regards simpler, but requires microscopic counting of parasites; the two assays are also shown to correlate quite well. This manuscript also offers an in depth characterization of genetic relatedness and sharing of drug resistance determinants among the studied Cambodian isolates and characterization of the stage specificity of piperazine. Overall, the manuscript is well-written and clear and the results are well-displayed and communicated. The manuscript offers a more in-depth characterization of the already well-described piperazine resistance genotype/phenotype in Cambodia. Some specific minor concerns are as follows.

We thank Reviewer 1 for the overall positive feedback on our manuscript and acknowledging that we provide a better way of measuring piperazine (PPQ) susceptibility in vitro than the recently described PPQ sensitivity assay (PSA).

However, we would like to point out that the bimodal response is a new phenotype or rather one that has been ignored as biologically relevant so far. While PPQ resistance has been observed in the field as recrudescence and in vitro as strange dose response curves, most investigators have excluded isolates from their studies as they couldn't calculate EC50 or removed data points as outliers¹. We are the first group that describes this specific phenotype in detail and we think that it has biological relevance to how PPQ resistance is achieved.

Plasmepsin II and III copy number variations (CNV) have been observed with PPQ resistance in the field, thus they serve as a surrogate marker of PPQ resistance. However, to date a direct link between plasmepsin II-III CNV and PPQ resistance has not been shown; and, we believe our data provides biological evidence that plasmepsin II-III CNV do indeed confer in vitro PPQ resistance.

Reviewer 2

[A caveat...I know no nothing about measuring viability in vitro or in conducting these types of dose/response assays...I have assumed the methods are appropriate and accurate but strongly recommend the ms is also reviewed by a specialist in this area]

The paper reports an interesting phenomenon i.e. bi-modality of a dose/kill curve (Figure 1). I enjoyed my first read but became increasingly concerned at subsequent reading as the phenomenon is so poorly described. In particular, the arguments often appear circular i.e. resistant parasites have bi-modal responses so any with bi-modal response are resistant... this becomes a big problem because the causal reader may think that when they talk about PPQ-resistant isolates that the resistance has been observed in patients i.e. as drug failures. Hopefully this is presentational but there are some clarifications that need to be made before publication i.e. as outlined in the "Majorpoints" below.

We thank Reviewer 2 for acknowledging that we identified a new phenotype for PPQ resistance. We understand that reviewer 2 has limited experience with drug susceptibility assays in vitro and we think some of

his/her concerns about the manuscript originate from this unfamiliarity.

Both reviewers encouraged us to provide a deeper discussion about the possible mechanism of this unusual bimodal response and we have added a paragraph addressing this in the discussion section.

We provide a point-by-point response to all the issues raised by the two reviewers and have grouped similar issues to be more concise.

Concerns raised by reviewer 1 and reviewer 2

1) **Abstract. The penultimate sentence is confusing. The disconnect between pfmdr1 copy number and PPQ phenotype is not surprising, and arguably need not be mentioned in the abstract. Rather, a broader summary statement addressing the observation that plasmepsin copy number appeared to explain piperazine resistance in most, but not all cases, would be more helpful.**

We thank the reviewer for pointing out this confusion and have edited the abstract to reflect the suggested changes.

(7) they should re-write the Abstract for the reason given above. For example they say “(AUC) correlating to PPQ resistance. A casual reader would assume this is a correlation to therapeutic outcome. In fact, AUC is their measure of PPQ resistance and the terms Bimodal and PPQR seem to be the same thing to i.e. PPQR is defined as a those isolates with a bimodal curve.

As stated above, the drug resistance is typically used to reflect a parasite's *in vitro* response to drug exposure. Clinical failure or clinical resistance is often used to reflect the clinical response to a drug and includes host factors and a variety of other considerations. However, to clarify for the broader reading audience, we have included “*in vitro*” in the abstract to avoid confusion.

New abstract:

Multi-drug resistant *Plasmodium falciparum* in Southeast Asia (SEA) endangers regional malaria elimination and threatens to spread to other malaria endemic areas. Understanding mechanisms of piperazine (PPQ) resistance is crucial for tracking its emergence and spread, and to develop effective strategies for overcoming it. We exposed Cambodian parasites to PPQ and observed a bimodal dose-response curve, with the area under the curve (AUC) correlating to *in vitro* PPQ resistance. *We find that increased copy number for plasmepsin II and plasmepsin III appeared to explain PPQ resistance in most, but not all cases.* A panel of isogenic subclones reinforces the importance of plasmepsin II – III copy number to PPQ resistance. These results confirm PPQ resistance in Cambodia, support the importance of plasmepsin II – III in mediating PPQ resistance, and suggest additional or alternate loci are important for PPQ resistance.

5) **Discussion. Paragraph beginning “As to the mechanism of PPQ resistance...”. The authors fail to note that increased plasmepsin copy number, which presumably affords greater plasmepsin activity, would be expected to lead to a greater degree of hemoglobin hydrolysis, thus more free heme, and thus GREATER activity of aminoquinolines such as chloroquine and PPQ, which act by preventing heme detoxification. Thus, as has been noted in many a hallway conversation at scientific meetings, the results, although clearly true, defy our understanding of aminoquinoline action and parasite biology. The last sentence of this paragraph (“Increased plasmepsin II-III copies could help parasites overcome inhibition of hemoglobin digestion by PPQ”) is misleading. Chloroquine and PPQ can inhibit hemoglobin digestion at very high concentrations, but their primary mode of antimalarial action is believed to be inhibition of heme detoxification, not inhibition of hemoglobin digestion. This is a complicated area and the authors need not dive into it fully, but their superficial discussion is unhelpful, and if they wish to try to explain the biology behind plasmepsin copy number and PPQ action, a more nuanced discussion is advised.**

We thank the reviewer for pointing out the nuances of possible mechanisms of PPQ resistance. We have rewritten the section in the discussion to reflect the concerns of the reviewer and to be more comprehensive in our discussion to reflect available data and how our findings may fit into this understanding of mechanism of

PPQ resistance.

(3) There is no discussion about the implications of stage-specific killing of the drug. PPQ is believed to interfere with haem metabolism (as noted on page 14) and I think the consensus is that it kills stages around 20 to 45 hours post-invasion when this metabolism occurs. The impact of this should be discussed. For example, figure 5 is based on 12 hour exposures... those parasites synchronised to ages 0 to 12 will only get a few hours killing (i.e. as they develop to age 20+ hours) how will this affect the dynamics??

We thank the reviewer for asking to connect the stage effects with potential mechanism. The purpose of Figure 5 was to address this issue about stage-specific effects, and we show all stages are affected by PPQ exposure except the very latest. The hemoglobin degradation process starts as soon as the parasites invade the RBC but is likely highest at the early trophozoite stage.

(8) The interesting part of this work is the observation of a bimodal distribution in sensitivity (Fig 1). How do they think this could arise? Could it be an artefact i.e. drug interfering with their assay?? They briefly discuss this on page 13 (“stress response pathways”) but I don’t see how that could generate a bimodal response. Has similar responses been noted to other drugs? If they have no idea why it arises, they shouldn’t be afraid to say this, but readers deserve some discussion as to why it could arise.

We thank the reviewer for noting that the bimodal response is an interesting finding. To ensure that it is not an artifact we used the hypoxanthine incorporation assay as well as the modified PSA where parasites survived better under higher drug concentrations. While there are no known antimalarial drugs that show a similar response, the Eagle effect has been observed in bacteria where high, but not low, doses of penicillin kill bacteria. We have included this information in the discussion. We have extended the discussion about this issue here:

*PQR parasites do not show a typical shift in EC_{50} values in conventional drug assays, but rather yield non-interpretable dose-response curves¹⁻⁴ that are often excluded from analysis. By increasing both the PPQ starting concentration and the number of concentrations tested, we identified a new and unusual PPQ phenotype whereby parasites survive better under higher PPQ concentrations than under some lower concentrations of drug. Other groups have likely missed this phenotype when using EC_{90} values or excluding data points as outliers^{1, 5, 6}. A collection of evidence, including use of [³H]-hypoxanthine incorporation and visualization of viable parasites, indicates that parasites exhibiting a bimodal response with high AUC values can survive these elevated PPQ concentrations (2 μ M) better than low concentrations (200nM). **This is reminiscent of the survival of certain bacterial strains to penicillin and other beta lactam antibacterials at higher concentrations, and has been termed the “Eagle effect”^{7, 8}.** AUC data correspond with previously reported PSA phenotypes and correlate with plasmepsin II – III copy number, demonstrating that the **bimodal response with high AUC is a valid phenotype for PPQ resistance.** The AUC assay is advantageous because it is easy, quick, and robust and does not require tight synchronization in vitro, or counting parasitemia by microscopy. AUC provides a broad dynamic range for PPQ response and may reveal intermediate phenotypes (AUC = 35 – 100) that may be useful for monitoring the emergence and spread of **PPQ resistance as many of these intermediate isolates had increased plasmepsin II – III copy numbers.** This bimodal PPQR phenotype suggests a unique mechanism of survival under high PPQ concentration. For example, perhaps PPQ induction of stress response pathways enables PPQR parasites to survive under high PPQ levels. Studies of this response can guide our understanding of PPQ resistance mechanisms to help explain how PPQR parasites survive drug exposure.*

And we have added an entire paragraph about the possible role of plasmepsins as suggested by reviewer 1 and 2:

*As to the mechanism of PPQ resistance involving plasmepsin II – III, PPQ is thought, like CQ, to **accumulate in the food vacuole (FV) and inhibit conversion of toxic heme moieties to non-toxic hemozoin crystals during hemoglobin digestion.** We have shown that **PPQ is acting on all stages of the parasite life-cycle but is least effective at the latest stages (Figure 5).** High-resolution live-cell imaging has shown that hemoglobin*

degradation is initiated early after invasion of the RBC by the parasite⁹. These authors suggest that cytosome-mediated endocytic events are the first step in the genesis of the hemoglobin-degrading apparatus of P. falciparum. Endocytosis of the host compartment is then followed by a process that concentrates the contents of the endocytic compartment and hemozoin crystals start to accumulate in late ring-stage parasites. It is therefore not surprising that PPQ is acting on early stages as well as later stages. There is less data available on hemoglobin degradation in late trophozoites/schizonts but it is possible that degradation slows down once parasites are converted into merozoites and are ready to egress. Plasmepsin II, plasmepsin III, and pfmdr1 are located in the food vacuole, with plasmepsin II and III directly involved in hemoglobin degradation generating small peptides and eventually amino acids for protein synthesis but also the toxic byproduct heme. Treatment of parasites with PPQ leads to a depletion of ribosomes and swelling of the FV with undigested hemoglobin vesicles and reduced hemozoin crystals¹⁰ as well as an accumulation of free heme¹¹. While the free heme is toxic for the parasite, inhibition of haemoglobin degradation will starve the parasite and increased plasmepsin II–III might help the parasite to maintain amino acid production when haemoglobin degradation is otherwise inhibited by PPQ. More experiments are needed to understand how plasmepsin II–III and PPQ influence haemoglobin degradation and the role of this process in PPQ resistance.

Reviewer 1 specific:

2) Introduction. First paragraph. ACTs are not used only to forestall resistance development, but also (and primarily) because artemisinin monotherapy has unacceptably high failure rates. At the end of the paragraph all references refer to piperazine resistance; at least one reference to support comments about mefloquine resistance would be helpful. Alternatively, mefloquine might not be mentioned, as this report is, of course, about PPQ.

Thank you for noting this omission, we have added references specifically for mefloquine resistance.

Partner drug resistance is also evident, including emergence of piperazine (PPQ) resistance in Cambodia and Viet Nam^{2, 12, 13}, and mefloquine (MEF) resistance on the Thailand-Myanmar border^{14, 15}.

3) Supplementary Figure 1b. The units of AUC should be noted.

We added the units for AUC, which is survival x PPQ concentration

4) Discussion. First sentence. It is unclear why the authors singled out use of DHA-PPQ for MDA. In fact, the drug is widely used for therapy, and its use in MDA remains mostly investigational. It would be more relevant to express concern about all areas where the drug is used, whether for treatment or MDA or IPT (so, consider omitting the last two words of the sentence).

We thank the reviewer for noting this, and have removed the last two words.

Reviewer 2 specific:

Major points

(1) Are the assays done under physiologically-plausibly conditions?? They need to put it into context. Their scale is in uM but many field studies report human concentration in ng/ml or mg/ml.... see example from Saunders et al at the end of this review. They should provide a conversion (most easily in the captions to Table 1) so readers can compare with concentration noted in patient's posttreatment (ng.ml, mg/ml). They use 72 hour exposure for Fig 1, whereas PPQ is believed to persist at active concentrations for weeks post-treatment (e.g. the Saunders et al graph below) so they must include an explicit, objective discussion of this.

We thank the reviewer for this question. Yes, the assay is done under physiologically relevant conditions, and we have included this in the manuscript.

We used uM rather than ng/ml as this is the more commonly used unit for in vitro assays. However, the point is well taken and we have added the following information about PPQ plasma levels in patients in the result section on page 5 after describing the shape of the curve:

“Interestingly, the first half of the second dose-response peak overlaps with PPQ plasma concentrations between 30-300ng/ml (or 56-560nM) found in patients after three days of PPQ treatment¹⁶.”

Please see response for (2) regarding assay details.

While we don't have a Table 1, possibly the reviewer meant Figure 1? We have left the units in the standard assay format, but have noted the comparison between the different units in the text (see above).

(2) Figure 1. they need to say that exposure was for 72 hours but, more importantly, that the parasites were “tightly synchronised 0-6 hour rings” (quote from the methods). Why were these ring stages selected over other stages (see also point 3 below)? They later look at different stages (Figure 5) but under different assay conditions (12 hour exposure)... so the question is whether this bimodal phenotype only occurs in rings or whether it occurs in other stages. This is major omission and needs to be discussed. I know how labs work and it may well be case that, in retrospect, they should have tested all the stages in the same way but they need to discuss this. I looked at the histogram on fig 5 and can see no clear signal of bimodality in later stages

- We thank the reviewer for asking about this assay. As stated above, 72-hour exposure is the classical way for drug testing in *P. falciparum*. Parasites are synchronized and exposed starting as ring stages to ensure that all stages are exposed to the drug. These assays are done in 384 well plates where changing of the media would be difficult to achieve. If parasites were exposed for longer times the no drug control would start to overgrow and falsify the results. Our lab has developed this assay in 2007 and we routinely use it in the lab regardless of the drug¹⁷. The difference between a growing control culture and a drug treated culture is best detectable at 72 hours where the control will be in the trophozoite/schizont stage that has the highest concentration DNA, which is measured by SYBR Green I and results in the signal-to-noise ratio.
- In addition, parasites circulating in the blood are mostly ring stage parasites. Our *in vitro* assay with ring stage parasites is therefore comparable to the *ex vivo* assays performed in the field.
- However, to clarify these points to the broader audience, we have included additional information in the results section on page 4:

One challenge to investigating PPQ resistance is defining the *in vitro* drug resistance phenotype for PPQ. Conventional drug susceptibility testing over **72 hours** to measure the half-maximal effective concentration (EC_{50}) resulted in incomplete parasite killing, with several parasite isolates surviving the highest drug concentration used. Using the PSA was not optimal since it was very labor-intensive and prone to microscopist bias. To gain better understanding of the biological response of parasites to PPQ and define PPQ resistance within this Cambodian parasite population, culture adapted clinical isolates were subjected to a modified dose-response approach that resulted in complete parasite killing. To achieve complete parasite killing for all isolates, the starting PPQ concentration was increased 100-fold (from 0.5 to 50 μ M) and the dilution series extended from 12 to 24 points. **Highly synchronized ring-stage parasites (0-6 hours) were exposed to these conditions for 72 hours.** Under these conditions, complete killing of all parasites at the highest drug concentration was achieved (Figure 1a, Supplementary Figure 1).

Regarding the question about testing in other stages: We actually did test if we observe a bimodal curve if we start at the trophozoite stage rather than the ring stage and while there is a trend, it is not as well defined likely due to how the assay is done.

We have shown that early stage parasites are more susceptible to PPQ than late stages (Figure 5). If we start with late stage parasites (36-48h) most of them will survive across any drug concentration in the first cycle (0-24h) and more parasites are viable in the second life cycle. The assay window between the no drug control and the assayed parasites decreases dramatically as there is less of an increase in DNA content even if parasites die as rings.

(4) the most interesting facet of this paper of the bi-modal sensitivity. It is not clear how consistent this is either over stages (see above) or over isolates. The caption to Figure 2 states “Bimodal growth was observed with increasing PPQ concentration for a subset of parasites”. This needs to be quantified...did the “subset” consist of a single isolate (KH0004_57) or many isolates? The answer is the latter but we need to go to SI figure 1 to find this information, which should be in the main text. Even then it is not easy to visualise individual curves. Did they assign something as being nonbimodal vs bimodal purely by eyeballing each curve?? I would have preferred a more objective criterion e.g “bimodality was defined as the presence of a second peak with 2 or more measures of $>xx\%$ growth and separated from the initial curve by at least two points with $\%$ growth less than yy ” or something similar. It was also not obvious to me (e.g. when they talk about “discordant” results) whether bi-modality also occurs in the absence of plasmepsin increased copy number and/or whether some isolates with increased plasmepsin copy number show unimodal profiles. This needs to be made explicit in the main text...they also need to be unambiguous when they talk about “discordant” results.

(6). I object to the statements in the first part of the discussion. I don’t believe they are supported by the data and need to be toned down. They state “we developed a new, robust method to quantify PPQ resistance” (I think this is tautological but I’ll let this pass). The point is they do not demonstrate it has any link to clinical resistance. I think they should be more circumspect. My understanding is that people have now shown the increasing plasmepsin copy number is a risk factor for therapeutic failure so they should say something like “Recent work (citations) has shown a link between increasing plasmepsin copy number and risk of drug failure, our results suggest mechanisms by which these phenotype may arise” or something along those lines. Similarly they state that their results “revealed that PPQR parasites exhibit a bimodal response to increasing PPQ concentration”... this does seem to be tautological... they define PPQR by bimodality, then claim bimodality identifies this phenotype. Overall, they just have to be more circumspect in discussing the implications of their study. Similarly on page 13 I do not follow/understand their argument about “bona fide phenotypes” and this needs to be clarified

We thank the reviewer for these points. There are numerous isolates with this phenotype and while this is shown in the supplemental figures, we are happy to bring it into the main text if the editor feels it is important. However, the AUC data from these curves are shown in Figure 3.

There seems to be some confusion about the use of the terms “bimodal” and “AUC” in the context of resistance. We thank the reviewer for pointing this out, and have explained below as well as added clarifying text in the manuscript.

- Drug resistance in malaria is a term commonly applied to parasites *in vitro* response to a given drug. We are using this term drug resistance accordingly. However, we added “*in vitro*” to clarify.
- A bimodal response was observed in almost all of the isolates. We don’t want to say that every parasite that has a bimodal response is resistant. However, as we can’t use the EC50 to quantify the survival response of the parasites we used the AUC defined as the area between the two local minima after the initial killing reflected in the dose-response data. The AUC gives us a value that quantifies the survival response of the parasites.
- All parasites have an AUC value and we show them in Figure 3. We choose to show average AUC for all isolates rather than the actual curves, as it is hard to visualize as the reviewer points out as well.
- In the absence of clinical data it is indeed impossible to show a direct correlation between a high AUC and clinical PPQ failure. However, we have compared our method to the PSA method that has been evaluated on field samples with clinical data and is currently used in the malaria field to assess PPQ resistance and show a good correlation between the two methods. We therefore conclude that parasites with high AUC (>100) are likely PPQ resistant and parasites with low AUC (<34) are likely sensitive.
- The term “*bona fide phenotype*” was used to indicate that AUC is an appropriate phenotype for PPQ resistance based upon the correlation with the PSA, and can be used to reflect this characteristic of parasites *in vitro*. We have removed the term “*bona fide*” to avoid confusion.

We have tried to address the issue and clarify the definition of resistant parasites on two separate places:

Investigation of responses to PPQ exposure revealed that several isolates showed a bimodal response to PPQ rather than a classical sigmoidal dose-response curve typical for antimalarial drugs (Figure 1a, Supplementary Figure 1). *While parasites were killed at the same rate between 1 and 10nM PPQ concentrations, there was a difference in how well parasites could survive under higher drug concentrations (0.1µM -10µM) resulting in a second peak of survival for a subset of the parasites (Figure 1a, Supplementary Fig. 1).*

and

We observed a range of AUC values (Supplementary Fig. 1) with a distribution that suggested three discrete groups (Supplementary Figure 1b), with the first breakpoint at ~35 *and the second at 100*. We compared AUC values for nine isolates to results obtained using the published PSA¹⁸ *to establish a resistance cutoff* (Supplementary Fig. 2). In contrast to the dose-response assay, the PSA exposes 0-3 hour ring-stage parasites to 200nM PPQ for 48 hours before determining parasitemia by microscopy. The relative growth of the drug treated parasite compared to non-drug treated control parasite culture after 72 hours was used to measure PPQ response with isolates demonstrating relative growth of >10% considered PPQ resistant¹⁸. Among these nine representative isolates, we found that PSA survival rates between 0 and 30% correlated well with AUC values (Spearman $r = 0.85$, $p = 0.0061$; Supplementary Fig. 2). *Isolates with AUC >100 were well above the >10% resistance cutoff defined in the PSA and we consider them PPQ resistant (PPQR). Isolates with AUC < 35 were all below the 10% cutoff and considered sensitive (PPQS).*

(5) The manuscript lack important details and quantification.

(i) on page 2 they say they tested 37 isolates from 157. On what criteria were these 37 chosen?

(iii) How long cultured?? The methods has a section called “culture-adaptation” but needs details. Given that subclones of some isolates differ in the copy number it would be nice to know how quickly the adaptation occurred. If it was very rapid then the copy number variation (CNV) may reflect what was in the patient, if very long then CNV may have arisen during the adaptation process.

The parasite population and handling is described more fully in our previously published paper¹⁹ that we reference.

- *Briefly, we obtained 157 isolates that were selected for their delayed clearance related to artemisinin resistance. We randomly adapted 65 of these parasites, and pursued a subset that exhibited the bimodal*

phenotype to PPQ in that original paper ($n = 28$). In pursuit of PPQ resistant isolates, we then used genetic relatedness to identify additional isolates ($n = 9$) for the phenotypic work in this manuscript.

- Initial parasites isolates were frozen at the first appearance of rings, and only allowed to grow for 2 weeks to create stocks for retrieval. For subsequent experiments, parasites were thawed and kept in culture for no longer than 2 months. The sub-cloning process takes about 1 month.
- To address concerns about changes in CNVs for plasmepsin II – III, we kept several isolates in continuous culture for over 6 months and monitored CNV levels across this time, and did not detect a decrease in the average copy numbers indicating that they are stable in vitro.
- We have included a clarification of this criterion in the material and methods section reading:

Culture-adaptation and maintenance of TRAC Parasites

All parasite samples were collected under protocols approved by ethical review boards in Cambodia, at Oxford University (OxTREC) and at the Harvard T.H. Chan School of Public Health. Culture-adaptation of parasites was accomplished by thawing cryopreserved material containing infected red blood cells (iRBCs) that had been mixed with glycerolyte. Parasites were maintained in fresh human blood (O+) and Hepes buffered RPMI media containing 10% O+ human serum (heat inactivated and pooled). Cultures were placed in modular incubators and gassed with 1%O₂/5% CO₂/balance N₂ gas and incubated with rotation (50 rpm) in a 37°C incubator.

Parasite lines were frozen within a few cycles (< 2 weeks) after parasites first appeared; and, never maintained for longer than 2 months in culture after these initial stocks were thawed. The 37 parasites used in this study were selected from among 157 previously adapted lines¹⁹. An initial sample subset ($n = 28$) was randomly identified, but an additional set ($n = 9$) was included based upon their barcode identity to KH001_053 to enrich for parasites that might have genetic variants that contribute to PPQ resistance.

(ii) A large proportion of isolates had increased plasmepsin copy number. Was there widespread PPQ failure at the collection sites (NB PPQ+DHA will fail, even if parasites are fully sensitive to DHA...its the partner drug that determines therapeutic outcome) . If there was no evidence of drug failure in patients it suggests the bi-modal phenotype may not yet be clinically relevant. I note they were collected in 2011 so that may be before PPQ resistance spread?

The reviewer makes a good point, however we unfortunately do not have clinical outcome data for these collected isolates and do not know if they failed or not. Witkowski et al²⁰ reported 9.4% recrudescence cases in Pailin in 2009 with numbers increasing to 19% in the next year. Recrudescence cases also increased in Pursat from 9.4% in 2010 to 17.1% in 2011 when the samples were collected.

(iv) I do not like the term “isogenic”. To me (and I suspect many readers) it implies an inbred line genetically engineered to contain only a single genetic difference. In the current case they mean sub-clones and I think they should stick to this term. They might argue that plasmepsin copy number is the only genetic difference between these sublicense (and hence they are isogenic) but they have not formally proved it, plus it is hard to understand why plasmepsin CNV should be the only variation within the isolate.

We appreciate the reviewers comment. According to Oxford Dictionary, isogenic refers to organisms “having the same or closely similar genotypes”, and does not require genetic manipulation to create these organisms. However, to clarify this point, we have stated a definition for isogenic and how we are using it in the manuscript.

WGS was performed on three of these subclones to confirm that identical molecular barcode genotypes represented isogenic lines—parasites that have the same or closely similar genotypes.

Minor points.

(A) Introduction...they imply that artemisinin (partial) resistance has led to the evolution of resistance to other drugs. This is a standard “SE Asia” argument but is controversial... all they need say is that PPQ-DHA is failing, and this is due to PPQ resistance (an effective partner drugs means its ACT will not fail).

We appreciate the reviewer's comment and have changed the language in the introduction to reflect this point. We now say:

Mutations in the *Pfkelch13* locus are associated with²¹ and confer ART resistance^{22, 23}. **Partner drug resistance is also evident, including emergence of piperavaquine (PPQ) resistance in Cambodia and Viet Nam^{2, 12, 13}, and mefloquine (MEF) resistance on the Thailand-Myanmar border^{14, 15}.**

Very minor points

- (i) **Page 4 "PSA" ...has the abbreviation been explained? Figure 3 cited before figure 2. Caption to Fig 3: have "qPCR" and "WGS" abbreviations been given. Page 8 has 'MEF' been defined**

We have checked to make sure all the abbreviations used in the manuscript are described fully at first use; and that the order of Figure citations is accurate. We are now not using abbreviations in the figure legends:

Figure 3. AUC correlates overall with plasmepsin II copy numbers but there are exceptions. Panel (a) shows CNVs for plasmepsin II (blue) and *pfmdr1* (red) that were estimated based upon read depth of **whole genome sequencing data and confirmed by quantitative real-time polymerase chain reaction.**

We added a clarifying statement in the text:

To better understand the biology of PPQ resistance, we analyzed a set of culture-adapted Cambodian parasites from the Tracking Resistance to Artemisinin Collaboration (TRAC), collected in Pursat and Pailin in 2011²⁴ **when the first cases of recrudescence were reported²⁰.**

References:

1. Chaorattanakawee, S. et al. Ex vivo piperavaquine resistance developed rapidly in *Plasmodium falciparum* isolates in northern Cambodia compared to Thailand. *Malar J* **15**, 519 (2016).
2. Leang, R. et al. Efficacy of dihydroartemisinin-piperavaquine for treatment of uncomplicated *Plasmodium falciparum* and *Plasmodium vivax* in Cambodia, 2008 to 2010. *Antimicrob Agents Chemother* **57**, 818-826 (2013).
3. Lim, P. et al. Ex vivo susceptibility of *Plasmodium falciparum* to antimalarial drugs in western, northern, and eastern Cambodia, 2011-2012: association with molecular markers. *Antimicrob Agents Chemother* **57**, 5277-5283 (2013).
4. Saunders, D.L. et al. Dihydroartemisinin-piperavaquine failure in Cambodia. *N Engl J Med* **371**, 484-485 (2014).
5. Agrawal, S. et al. Association of a Novel Mutation in the *Plasmodium falciparum* Chloroquine Resistance Transporter With Decreased Piperavaquine Sensitivity. *J Infect Dis* **216**, 468-476 (2017).
6. Parobek, C.M. et al. Partner-Drug Resistance and Population Substructuring of Artemisinin-Resistant *Plasmodium falciparum* in Cambodia. *Genome Biol Evol* **9**, 1673-1686 (2017).
7. Holm, S.E., Tornqvist, I.O. & Cars, O. Paradoxical effects of antibiotics. *Scandinavian journal of infectious diseases. Supplementum* **74**, 113-117 (1990).
8. Eagle, H. & Musselman, A.D. The rate of bactericidal action of penicillin in vitro as a function of its concentration, and its paradoxically reduced activity at high concentrations against certain organisms. *The Journal of experimental medicine* **88**, 99-131 (1948).
9. Abu Bakar, N., Klonis, N., Hanssen, E., Chan, C. & Tilley, L. Digestive-vacuole genesis and endocytic processes in the early intraerythrocytic stages of *Plasmodium falciparum*. *Journal of cell science* **123**, 441-450 (2010).

10. Sachanonta, N. et al. Ultrastructural and real-time microscopic changes in *P. falciparum*-infected red blood cells following treatment with antimalarial drugs. *Ultrastructural pathology* **35**, 214-225 (2011).
11. Dhingra, S.K. et al. A Variant PfCRT Isoform Can Contribute to Plasmodium falciparum Resistance to the First-Line Partner Drug Piperaquine. *mBio* **8** (2017).
12. Thanh, N.V. et al. Rapid decline in the susceptibility of Plasmodium falciparum to dihydroartemisinin-piperaquine in the south of Vietnam. *Malar J* **16**, 27 (2017).
13. Amaratunga, C. et al. Dihydroartemisinin-piperaquine resistance in Plasmodium falciparum malaria in Cambodia: a multisite prospective cohort study. *Lancet Infect Dis* **16**, 357-365 (2016).
14. Shanks, G.D. 1993 Sir Henry Wellcome Medal and Prize recipient. The rise and fall of mefloquine as an antimalarial drug in South East Asia. *Military medicine* **159**, 275-281 (1994).
15. Thimasarn, K. et al. In vivo study of the response of Plasmodium falciparum to standard mefloquine/sulfadoxine/pyrimethamine (MSP) treatment among gem miners returning from Cambodia. *The Southeast Asian journal of tropical medicine and public health* **26**, 204-212 (1995).
16. Tarning, J. et al. Population pharmacokinetics of dihydroartemisinin and piperaquine in pregnant and nonpregnant women with uncomplicated malaria. *Antimicrob Agents Chemother* **56**, 1997-2007 (2012).
17. Baniecki, M.L., Wirth, D.F. & Clardy, J. High-throughput Plasmodium falciparum growth assay for malaria drug discovery. *Antimicrob Agents Chemother* **51**, 716-723 (2007).
18. Duru, V. et al. Plasmodium falciparum dihydroartemisinin-piperaquine failures in Cambodia are associated with mutant K13 parasites presenting high survival rates in novel piperaquine in vitro assays: retrospective and prospective investigations. *BMC medicine* **13**, 305 (2015).
19. Mukherjee, A. et al. Artemisinin resistance without pfcy8 mutations in Plasmodium falciparum isolates from Cambodia. *Malar J* **16**, 195 (2017).
20. Witkowski, B. et al. A surrogate marker of piperaquine-resistant Plasmodium falciparum malaria: a phenotype-genotype association study. *Lancet Infect Dis* (2016).
21. Ariey, F. et al. A molecular marker of artemisinin-resistant Plasmodium falciparum malaria. *Nature* **505**, 50-55 (2014).
22. Ghorbal, M. et al. Genome editing in the human malaria parasite Plasmodium falciparum using the CRISPR-Cas9 system. *Nat Biotechnol* **32**, 819-821 (2014).
23. Straimer, J. et al. Site-specific genome editing in Plasmodium falciparum using engineered zinc-finger nucleases. *Nat Methods* **9**, 993-998 (2012).
24. Ashley, E.A. et al. Spread of artemisinin resistance in Plasmodium falciparum malaria. *N Engl J Med* **371**, 411-423 (2014).

REVIEWERS' COMMENTS:

Reviewer #2 (Remarks to the Author):

Most of my more technical questions have been addressed.

I remain concerned that it will mislead the casual reader as the authors do not make a clear distinction between resistance in vivo (which they investigate) and resistance in practice (i.e. clinical failures) which they lack the data to address. The Introduction to the paper (and the first two sentences of the abstract) discusses their work's importance in clinical resistance patterns in SE Asia and I consider the Authors to be disingenuous on Page 6 of their response where they state

Drug resistance in malaria is a term commonly applied to parasites in vitro response to a given drug. We are using this term drug resistance accordingly. However, we added "in vitro" to clarify.

I don't think adding the words "in vitro" once in the abstract is a serious attempt to draw this distinction and to avoid confusion.

In fact, Lines 129 to 130 tell us "The relative growth of the drug treated parasite compared to non-drug treated control parasite culture after 72 hours was used to measure PPQ response with isolates demonstrating relative growth of >10% considered PPQ resistant"

I strongly suggest they add (or the Editor insists upon) a final sentence to the abstract that firmly and unambiguously draws this distinction e.g. "We conjecture that these factors producing resistance in vitro may also drive clinical failures in vivo".

Response to Reviewer Comments:

We appreciate the request from Reviewer #2 to be more explicit about the distinction between resistance *in vitro* (which we investigate) and resistance in practice (i.e., clinical failures).

To highlight this distinction and avoid confusion that may mislead the casual reader, we have included the following changes or additions to the manuscript:

1. We have converted the use of “drug resistance” to “enhanced survival under PPQ exposure” whenever possible.
2. We added the following last sentence to the abstract as suggested by the reviewer: “We conjecture that factors producing increased parasite survival under PPQ exposure *in vitro* may drive clinical PPQ failures in the field.”
3. We added an additional sentence to the last paragraph of the introduction: “Characterizing the survival of cultured parasites exposed to PPQ reveals changes that inform possible mechanisms of clinical PPQ resistance evident in Cambodia.”
4. We added an additional sentence to the first paragraph of the discussion: “To explore mechanisms of PPQ resistance we leveraged cultured parasites from Cambodia, where clinical PPQ resistance has recently emerged, to identify phenotypic and genotypic characteristics of PPQ response among these parasites that may help elucidate possible mechanisms of clinical PPQ resistance observed among patients.”